# Effects of Ketogenic Diet on Increased Ethanol Consumption Induced by Social Stress in Female Mice

**DOI:** 10.3390/nu16172814

**Published:** 2024-08-23

**Authors:** Laura Torres-Rubio, Marina D. Reguilón, Susana Mellado, María Pascual, Marta Rodríguez-Arias

**Affiliations:** 1Unit of Research Psychobiology of Drug Dependence, Department of Psychobiology, Facultad de Psicología, Universitat de Valencia, Avda. Blasco Ibáñez, 21, 46010 Valencia, Spain; laura.torres@uv.es (L.T.-R.); marina.reguilon@uv.es (M.D.R.); 2Department of Physiology, School of Medicine, Universitat de Valencia, Avda. Blasco Ibáñez, 15, 46010 Valencia, Spain; susana.mellado@uv.es (S.M.); maria.pascual@uv.es (M.P.)

**Keywords:** ketogenic diet, social stress, ethanol, female, mice

## Abstract

Stress is a critical factor in the development of mental disorders such as addiction, underscoring the importance of stress resilience strategies. While the ketogenic diet (KD) has shown efficacy in reducing alcohol consumption in male mice without cognitive impairment, its impact on the stress response and addiction development, especially in females, remains unclear. This study examined the KD’s effect on increasing ethanol intake due to vicarious social defeat (VSD) in female mice. Sixty-four female OF1 mice were divided into two dietary groups: standard diet (*n* = 32) and KD (*n* = 32). These were further split based on exposure to four VSD or exploration sessions, creating four groups: EXP-STD (*n* = 16), VSD-STD (*n* = 16), EXP-KD (*n* = 16), and VSD-KD (*n* = 16). KD-fed mice maintained ketosis from adolescence until the fourth VSD/EXP session, after which they switched to a standard diet. The Social Interaction Test was performed 24 h after the last VSD session. Three weeks post-VSD, the Drinking in the Dark test and Oral Ethanol Self-Administration assessed ethanol consumption. The results showed that the KD blocked the increase in ethanol consumption induced by VSD in females. Moreover, among other changes, the KD increased the expression of the ADORA1 and CNR1 genes, which are associated with mechanisms modulating neurotransmission. Our results point to the KD as a useful tool to increase resilience to social stress in female mice.

## 1. Introduction

Social stress is a critical factor for neurological and behavioral alterations that may lead to the development of mental disorders, playing a crucial role in the onset of substance-use disorders [1,2,3,4]. Drugs of abuse and stress response act upon common neurobiological pathways, among which is the reward system [5,6]. Stressful experiences alter the reward system’s response, influencing the consumption and compulsive seeking of abused substances, and contributing to the consolidation of addictive behavior [1,6,7,8].

In preclinical studies, the Social Defeat (SD) paradigm is frequently employed as a stress inducer in male mice. In this procedure, the experimental mouse is subjected to defeat by an aggressive and dominant male from the territory [6,7]. Animals exposed to SD exhibited an increase in anxiety and depression-related behaviors, along with elevated corticosterone blood levels, indicating the validity of the protocol as a stressful experience [9,10,11]. Previous studies using a repeated and intermittent SD paradigm have demonstrated that it induces an increase in consumption and motivation for cocaine and ethanol [1,12,13,14]. Exposure to SD has also been associated with an increase in the neuroinflammatory response and disruption of the dopaminergic system. Animals exposed to SD showed elevated levels of proinflammatory cytokines such as interleukin-6 (IL-6) and interleukin-1β (IL-1β), and chemokines such as fractalkine (CX3CL1) and C-X-C motif chemokine 12 (CXCL12), along with a decrease in the affinity and levels of the gene encoding the dopamine 1 receptors (DrD1) in the prefrontal cortex, amygdala, and hippocampus [1,15,16,17]. However, in studies involving female mice, their inherent low predisposition to being attacked by a conspecific of any gender complicates the use of the SD paradigm. Faced with this challenge, we have developed an adaptation of the Vicarious Social Defeat (VSD) paradigm [2,18], in which the female mouse is indirectly exposed to the social defeat experience. In this protocol, while a male intruder is defeated by an aggressive and dominant male over the territory, the female is a separated witness of the scene behind a barrier, allowing for visual, auditory, and olfactory cues perception without any direct physical involvement in the encounter. Once the male intruder’s defeat experience is concluded, the female remains in the dominant male’s cage, but physically separated for 24 h, enhancing the stress experience. Under this procedure, female mice displayed long-lasting heightened conditioned rewarding effects of cocaine and increased ethanol consumption [2,18]. They also exhibited increased social avoidance behaviors, anxiety, anhedonia, decreased prepulse inhibition (PPI), and higher levels of blood corticosterone, along with an elevation of proinflammatory markers such as IL-6 [2,18,19].

Given the significant impact that social stressors exert at the neurobiological and behavioral levels, coupled with their ubiquitous presence in our daily lives, it becomes imperative to develop strategies focused on mitigating the impact of stress experiences on the development of addictive behaviors. In recent years, there has been a notable surge in research focused on nutritional interventions as a therapeutic approach. The type of diet has been correlated with the synthesis and metabolism of various neurotransmitters, thereby exerting a significant influence on human emotions and brain functions [20], as well as contributing to the pathogenesis of mental disorders such as depression [21]. In the field of drug addiction, findings from animal models further support the association between diet and drug use. Specifically, evidence from our laboratory and others suggests that a high-fat diet (HFD) may affect cocaine and ethanol consumption [22,23,24,25,26]. In those studies, animals exposed to binge episodes of HFD demonstrated heightened sensitivity to the reinforcing effects of a subthreshold dose of cocaine and increased ethanol intake, leading to increased self-administration of both substances [23,24]. Notably, HFD also exhibited a role as an alternative reinforcer to drugs [22,27].

The ketogenic diet (KD) is a dietary regimen characterized by high fat intake, very low carbohydrate levels, and balanced protein intake that induces a metabolic shift toward utilizing ketone bodies as the primary energy source, replacing glucose [28]. It has demonstrated therapeutic efficacy in various neurological disorders, including epilepsy [29,30], Alzheimer’s [31], and Parkinson’s disease [32]. This protective efficacy is attributed to the diet’s antioxidant and anti-inflammatory effects [32,33,34]. Research on the role of the KD in relation to substance abuse is limited. Preclinical studies have demonstrated that the administration of this diet significantly reduces consumption and withdrawal symptoms of cocaine and ethanol compared to groups fed a standard diet [35,36,37,38,39,40], without altering the behavioral profile [41]. Biochemically, the KD was shown to increase the expression of the gene encoding the adenosine A1 receptor (ADORA1) without affecting the gene encoding the adenosine A2 receptor (ADORA2) and dopaminergic genes. However, this expression profile was reversed following exposure to ethanol [35]. These findings illustrate the correlation between the KD and the mechanisms influencing the brain’s reward system, suggesting the potential therapeutic utility of the KD in addressing substance abuse. The new therapeutic approaches in the utilization of the KD focus on pathologies related to inflammation and neuroinflammation [42], leading us to hypothesize that the KD could be effective in reducing neuroinflammation induced by stress experiences. However, it is essential to note that all these findings pertain exclusively to males.

Due to the absence of prior studies associating the KD with the ability to modulate the stress experience, the objective of this study was to evaluate the capacity of the KD as a modulator of the impact of stress experience and subsequent ethanol consumption in female mice. With this aim, KD exposure was conducted prior to and during stress experience, spanning from adolescence to the day of the last VSD. Thus, we will be able to determine the long-term effects of ethanol consumption induced by social stress in female mice under a ketosis state throughout the developing brain periods of adolescence. Furthermore, a gene expression analysis was performed at the end of the experimental procedure to investigate the consequences of long-term diet-induced modifications in reward-associated receptors, indicative markers of the neuroinflammatory immune response and neurotrophic factors.

## 2. Materials and Methods

### 2.1. Subjects

In this study, a total of 64 female OF1 mice obtained from Charles River (France) were used. The mice arrived at the laboratory on the 21st post-natal day (PND) and were housed four at a time in cages (28 cm × 28 cm × 14.5 cm) under standard feeding conditions until PND 25 when experimental feeding conditions began. For the VSD procedure, one group of adult male mice (*n* = 21) was individually housed for at least one month in cages (21 cm × 32 cm × 20 cm) to enhance their aggressiveness and to be used as aggressive opponents. Another group of adult male mice (*n* = 20) were housed in groups of four under standard conditions and served as intruders. All mice were kept in a controlled laboratory environment with a constant temperature (21 ± 2 °C) and a standard 12-h light/dark cycle. Food (standard or ketogenic diet) and water were provided ad libitum to the mice except during behavioral tests.

All procedures were conducted in full compliance with national, regional, and local laws and regulations, as well as Directive 2010/63/EU of the European Parliament and the Council regarding the protection of animals used for scientific purposes. Additionally, the Committee for the Use and Care of Animals at the University of Valencia approved the study under reference number 2022/VSC/PEA/0007.

### 2.2. Experimental Design

Female mice were randomly assigned to different groups based on diet type and exposure to VSD, resulting in four groups: (1) Standard diet without stress exposure (exploration; STD-EXP; *n* = 16), (2) Standard diet exposed to vicarious social defeat (STD-VSD; *n* = 16), (3) Ketogenic diet without stress exposure (KD-EXP; *n* = 16), and (4) Ketogenic diet exposed to vicarious social defeat (KD-VSD; *n* = 16). Mice in the KD groups started the experimental feeding conditions on PND 25 and continued on the KD until 24 h after the last VSD episode, then switched to STD on PND 65 until the end of the experiment. Mice in the STD groups remained on this diet throughout the study. VSD began on PND 55 and ended on PND 65. Twenty-four hours after the last VSD, when the female mice returned to their cages, they underwent the Social Interaction Test (SIT) to assess social avoidance and classify subjects as Resilient or Susceptible to VSD-induced depressive symptoms. Three weeks later, all female mice began the Drinking in the Dark (DID) procedure, which consisted of four sessions from PND 88 to PND 91 to habituate the subjects to ethanol. The following week, the Oral Ethanol Self-Administration (SA) procedure began, lasting from PND 95 to PND 135. On PND 136, brain tissue samples from the striatum and the hippocampus were collected for further analysis. A detailed description of the experimental procedure is provided in Figure 1.

### 2.3. Apparatus and Procedure

#### 2.3.1. Feeding Conditions, Body Weight, Kcal Intake, and Ketosis State

Two types of diets were used: STD and KD, both obtained from Envigo Teklad Diets (Barcelona, Spain). The STD consisted of the Teklad Global Diet 2014, composed of 13 kcal% fat, 67 kcal% carbohydrates, and 20 kcal% protein, with a caloric density of 2.9 kcal/g. The KD (TD.96355) contained 90.5 kcal% fat, 0.3 kcal% carbohydrates, and 9.1 kcal% protein, with a caloric density of 6.7 kcal/g.

Body weight was recorded twice a week, and caloric intake was monitored three times a week for all subjects. Throughout the VSD procedure, female mice on a KD maintained the same dietary regimen while in the resident cage. When the female mice returned to their home cage, the leftover food was also weighed for both the STD and KD groups.

To assess the animals’ ketosis state, plasma β-hydroxybutyrate (βHB) levels were measured once a week by collecting blood samples from the tail vein using an On Call GK Dual monitor with ketone test strips (ACON Laboratories, Inc., San Diego, CA, USA).

#### 2.3.2. Drug Treatment

For DID and SA administration, a 20% (*v*/*v*) solution of absolute ethanol (Merck; Madrid, Spain) in water was used.

#### 2.3.3. Vicarious Social Defeat (VSD)

In this study, female OF1 mice were exposed to VSD as a stressor following the protocol described by González-Portillo et al. [18]. During VSD sessions, female mice indirectly faced the defeat of their male OF1 counterparts through non-physical sensory stimuli, including visual, olfactory, and chemosensory cues related to the stressful confrontation. In each VSD session, an intruder male mouse was introduced into the cage of the aggressive resident male, while the female occupied an adjacent compartment separated by a grid. This setup facilitated incidental exposure to aggressive interactions, restricted to visual, olfactory, and auditory signals. VSD sessions lasted 15 min, with a total of four sessions spaced 72 h apart. After each VSD session, the female remained in the resident’s cage for 24 h, with a perforated Plexiglas wall (31 cm × 18 cm × 0.6 cm) as a barrier, allowing for sensory exposure while preventing direct physical interaction with the aggressive male mouse. After the 24 h interval, the female was returned to her home cage with her companions until the next VSD session. In contrast, the control group of females participated in four exploration sessions spaced 72 h apart, where each female mouse was placed in a different cage from her home cage and allowed to explore for 15 min daily. After each exploration session, the females were returned to their home cage.

#### 2.3.4. Social Interaction Test (SIT)

The Social Interaction Test (SIT) was carried out 24 h after the 4th VSD session, 2 h after the female mice returned to their cage after spending 24 h in the resident mouse cage. The SIT took place during the dark cycle and in a different experimental environment than the confrontational sessions. Following the protocol described to Ródenas-González et al. [2], female OF1 mice were acclimated for one hour in a quiet, dimly lit room before the test. In the SIT, each female mouse was placed in a black Plexiglas arena (30 cm sides, 35 cm height) with a perforated and transparent Plexiglas cage (10 cm × 6.5 cm × 35 cm) in one of its walls. Behavior was recorded using a video camera (EthoVision XT 11, 50 fps) placed above the arena for two 600 s sessions: the first with an empty Plexiglas cage (object session) and the second with an unfamiliar male mouse inside the Plexiglas cage (social session). The experimental mouse was temporarily removed for two minutes between object and social sessions. To minimize odor cues, the arena was cleaned after each complete SIT session. EthoVision XT 11 software tracked mice movement and assessed arena occupancy, including time spent in the interaction zone and corners. The interaction zone, representing social preference or avoidance, was a 6.5 cm wide area around the Plexiglas cage. Social withdrawal behavior was quantified using the SIT ratio, calculated by comparing time spent in the interaction zone with and without a social target. Among defeated mice, a SIT ratio below 1 indicated social avoidance and the mouse was classified as susceptible. Based on the typical control male mouse behavior, a SIT ratio of 1 or higher indicated social engagement, and the mouse was classified as resilient [43,44].

#### 2.3.5. Drinking in the Dark Test (DID)

Following the protocol established by Rhodes et al. [45], female mice were placed in individual cages equipped with a graduated suction tube, which had a ball bearing at the end to prevent any potential spills.

The test spanned four days and was employed as ethanol habituation for the oral ethanol self-administration procedure. On the first day of testing, the mice stayed in their individual cages for 2 h with a graduated suction tube containing 5 mL of a 20% (*v*/*v*) ethanol solution mixed with water. Afterward, they were returned to their group cages, where they had unrestricted access to food and water. Days 2 and 3 followed the same conditions as the initial testing day. On day 4, the animals were placed in their individual cages for 4 h with an ethanol solution identical to the previous test days.

A fresh ethanol solution was prepared each day, and the amount of fluid consumed was measured immediately after each session using a millimeter measuring cylinder.

#### 2.3.6. Oral Ethanol Self-Administration (SA)

This procedure is based on the method employed by Ródenas-González et al. [2]. Oral ethanol self-administration (SA) was conducted in 10 modular operant chambers (MED Associated Inc., Georgia, VT, USA). The software package SOF-735M (Cibertec, SA, Madrid, Spain) controlled stimulus and fluid delivery while also recording operant responses. These chambers were housed within noise-isolation boxes and equipped with a chamber light, two nose-poke holes, a receptacle for liquid delivery, a syringe pump, a stimulus light, and a buzzer. Active nose-pokes resulted in the delivery of 20 µL of fluid, accompanied by a 0.5 s stimulus light and a 0.5 s buzzer beep, followed by a 6 s time-out period. Inactive nose-pokes did not yield any consequences.

To assess the impact of VSD on the acquisition of oral ethanol self-administration (SA), animals underwent a three-phase experiment: training, fixed ratio 1 (FR1), and progressive ratio (PR), using a 20% ethanol concentration. During the training phase (18 days), mice were conditioned to respond to the active nose-poke in order to receive a 20 µL of 20% (*v*/*v*) ethanol reinforcement. No food or water deprivation was implemented in this protocol. In the subsequent FR1 phase (10 days), we aimed to evaluate the number of responses on the active nose-poke, the intake of 20% ethanol (*v*/*v*), and the motivation to consume ethanol. Following each session, the remaining alcohol in the receptacle was collected and quantified using a micropipette. To achieve this objective, the number of effective responses and ethanol consumption (in µL) were measured under a fixed ratio 1 (FR1) for ten consecutive daily sessions.

The PR phase (1 day) was conducted to determine the breaking point for each animal, defined as the maximum number of nose-pokes an animal could perform to earn one reinforcement. The response requirement to attain reinforcements increased following this series: 1-2-3-5-12-18-27-40-60-90-135-200-300-450-675-1000. To assess motivation toward ethanol consumption, the breaking point was calculated for each animal as the maximum number of consecutive responses needed to obtain one reinforcement based on the scale. For instance, if an animal activated the nose-poke 68 times, it would mean that it could consecutively respond a maximum of 27 times for one reinforcement. Consequently, the breaking point value for this particular animal would be 27. All sessions lasted for one hour, except for the PR session, which extended to two hours.

#### 2.3.7. Determination of Plasma Corticosterone (ELISA)

Blood sampling for corticosterone determination involved the tail-nick procedure. During this procedure, the animal was swathed in a cloth, and a 2 mm incision was made at the tail artery’s tip. Subsequently, the tail was gently massaged until 50 µL of blood was collected into an ice-cold Microvette CB 300 capillary tube (Sarstedt, Nümbrecht, Germany). The blood samples were kept chilled, and plasma was separated from whole blood through centrifugation (5 min, 5000× *g*) and transferred to sterile 2 mL microcentrifuge tubes. These plasma samples were stored at freezing temperatures until corticosterone determination.

On the day of the assay, the samples were diluted at a ratio of approximately 1:40 using the Steroid Displacement Reagent mix provided with the kit. Corticosterone levels in the diluted plasma were subsequently analyzed using a corticosterone EIA kit (Enzo Life Sciences, Farmingdale, NY, USA, Catalog No. ADI-900-097, 96-Well kit), following the manufacturer’s instructions. This analysis was performed using an iMark microplate reader (Bio-Rad, Hercules, CA, USA) and Microplate Manager 6.2 software. Optical density was measured at 405 nm, with a correction at 590 nm. The test’s sensitivity is 0.2.

#### 2.3.8. Tissue Sampling and Biochemical Analyses

Mice were euthanized by cervical dislocation. The striatum and hippocampus were meticulously dissected in accordance with the Paxinos and Franklin atlas [39] using a coronal brain matrix. These tissue samples were preserved at −80 °C until the RNA extraction.

#### 2.3.9. RNA Isolation, Reverse Transcription, and Quantitative RT-PCR

Striata and hippocampi were lysed in a Tri-Reagent solution (Sigma-Aldrich, Madrid, Spain), and the total RNA extraction was performed following the manufacturer’s instructions.

Subsequently, mRNA was reverse-transcribed using the High-Capacity cDNA Reverse Transcription Kit (Applied Biosystems, Waltham, MA, USA). Amplification of both the target and housekeeping genes was carried out using the AceQ^®^ qPCR SYBR Green Master Mix (NeoBiotech, Nanterre, France) and TaqManTM Fast Advanced Master Mix (Applied Biosystems), following the manufacturer’s instructions in a QuantStudio^TM^ 5 Real-Time PCR System (Applied Biosystems).

The mRNA level of housekeeping genes (cyclophilin A and β-glucuronidase) was used as an internal control for the normalization of the analyzed genes. All the RT-qPCR runs included non-template controls (NTCs). Experiments were performed in triplicates. Quantification of expression (fold change) from the Cq data was calculated by the QuanStudioTM Design & Analysis Software v1.5.2 (Applied Biosystems). The following genes were analyzed: Dopamine receptor D1 and D2 (DrD1, DrD2), adenosine A1 and A2 receptor (ADORA1, ADORA2), Cannabinoid receptor 1 (CNR1), Corticotropin-releasing hormone receptor 1 (CRHR1), Opioid receptor mu-1 (OPRM), Brain-Derived Neurotrophic Factor (BDNF), Tropomyosin receptor kinase B (TrkB), Interleukin-6 (IL-6), Interleukin-1β (IL-1β), Interleukin-10 (IL-10), and Toll-like receptor-4 (TLR4). Details of the nucleotide sequences and the assay codes of the used primers are detailed in Table 1 and Table 2.

### 2.4. Statistical Analysis

For the data of body weight and kcal intake, a mixed two-way analysis of variance (ANOVA) was used with two between-subjects variables: Stress (3 levels for bodyweight data: Non-stressed, VSD-Resilient, and VSD-Susceptible; 2 levels for kcal intake data: Non-stressed, and VSD) and Diet (STD vs. KD). Additionally, a within-subjects variable (weeks) was included, with 16 levels for body weight, 6 levels for kcal intake during the ketogenic diet period, and 10 levels for kcal intake when all mice were on the standard diet. The data of the BOH levels to measure ketosis state between the KD and STD groups in the sixth week were analyzed using a two-way ANOVA, with a between-subjects variable Stress (Non-stressed vs. VSD-Resilient vs. VSD-Susceptible) and Diet (STD vs. KD). Additionally, ketosis state data within the KD-fed groups throughout the experiment’s duration were analyzed by a mixed two-way ANOVA with the between-subjects variable Stress (Non-stressed vs. VSD-Resilient vs. VSD-Susceptible) and the within-subjects variable Week (6 levels).

The corticosterone levels data were analyzed by a mixed two-way ANOVA with two between-subjects variables: Stress (Non-stressed vs. VSD-Resilient vs. VSD-Susceptible) and Diet (STD vs. KD). Additionally, a within-subjects variable, VSD session, was included with two levels (1st VSD and 4th VSD).

Behavioral data from the SIT and gene expression data were analyzed using a two-way ANOVA with two between-subjects variables: Stress (3 levels for SIT Non-stressed, VSD-Resilient, and VSD-Susceptible; 2 levels for gene expression data: Non-stressed vs. VSD) and Diet (STD vs. KD). For DID and ethanol self-administration acquisition, a mixed two-way ANOVA was conducted, with the two between-subjects variables: Stress and Diet, each with three and two levels, respectively (Non-stressed vs. VSD-Resilient vs. VSD-Susceptible/STD vs. KD), and a within-subjects variable, Day with four or ten levels for DID and FR1, respectively. The impact of VSD on breaking point values (BPV), ethanol consumption, and effective responses during PR was analyzed by a two-way ANOVA with two between-subjects variables: Stress and Diet.

The results are presented as mean ± SEM, and statistical significance was established at a *p*-value < 0.05. All statistical analyses were carried out using SPSS v26, and Bonferroni tests were used for post hoc comparisons.

## 3. Results

### 3.1. Body Weight

The ANOVA for body weight showed a significant effect of the variable Week (F (15, 870) = 404.206; *p* < 0.000) and the interaction between Week × Stress (F (30, 870) = 3.359; *p* < 0.001), Week × Diet (F (15, 870) = 10.098; *p* < 0.001) and Week × Stress × Diet (F (30, 870) = 2.322; *p* < 0.001). The weight increase was attributable to the progressive development of female mice from adolescence to adulthood (*p* < 0.001 in all cases). Stressed females, both the resilient and susceptible profiles, irrelevant of the food condition, presented higher body weights in the fourth week compared to the non-stressed groups (*p* < 0.05 in both cases). Post hoc analyses revealed that, within the non-stressed groups, females on the KD exhibited higher body weights from the 2nd to the 7th week in comparison to those fed on the STD diet (*p* < 0.01 for the 2nd, 3rd, and 5th weeks, and *p* < 0.05 for the 4th, 6th, and 7th weeks). However, in the stressed groups, the differences in weight were less pronounced. In the Resilient groups, females on the KD exhibited higher body weights than those fed on the STD diet in the 2nd (*p* < 0.001), 3rd, and 6th (*p* < 0.05 in both cases) weeks. In the Susceptible groups, females on the KD only showed higher body weights than those fed on the STD diet in the 2nd week (*p* < 0.05) (see Figure 2).

### 3.2. The Kcal Intake

The ANOVA for kcal intake during the KD period (Weeks 1–5) showed a significant effect of the variable Stress (F (1, 12) = 6.066; *p* < 0.03) and Diet (F (1, 12) = 348.403; *p* < 0.001), Week (F (4, 48) = 32.775; *p* < 0.001) and the interactions Week × Stress (F (4, 48) = 4.891; *p* < 0.002) and Week × Diet (F (4, 48) = 3.090; *p* < 0.024). The stressed groups consumed less kcal than non-stressed groups (*p* < 0.05) and groups fed on the KD consumed more kcal than the groups fed on STD (*p* < 0.001), which probably is due to the caloric difference between the two types of diet. Mice significantly decreased their kcal consumption on the 5th week compared to the previous weeks (*p* < 0.05 in all cases), but in addition, defeated females significantly decreased their kcal consumption on the 5th week compared to non-stressed mice (*p* < 0.001) when the VSD sessions started.

On the standard diet period, the ANOVA for the kcal intake showed a significant effect of the variable Week (F (9, 108) = 93.666; *p* < 0.001) and the interactions Week × Stress (F (9, 108) = 2.651; *p* < 0.008), Week × Diet (F (9, 108) = 6.939; *p* < 0.001) and Week × Stress × Diet (F (9, 108) = 2.216; *p* < 0.026). The mice significantly reduced their caloric intake in the tenth week compared to the other weeks (*p* < 0.001 in all cases), an effect attributed to the development of DID procedure during that period. Post hoc analyses found that VSD females consumed more kcal than the non-stressed females during the 8th and the 9th week (*p* < 0.05). Furthermore, the groups fed on the KD showed significantly lower kcal consumption compared to the STD-fed groups in the 7th (*p* < 0.01) and 8th week (*p* < 0.05).

Week 6 was eliminated from the analyses as it was a transition week between the KD and the STD. The groups on the KD switched to the standard diet mid-week, so this week shows the effect of both types of diets (see Figure 2).

### 3.3. Ketosis State

For the βHB levels in the 6th week, the ANOVA found a significant effect of the variable Diet (F (1, 58) = 77.310; *p* < 0.001), meaning that KD animals had significantly higher levels of βHB than mice fed on standard diet (*p* < 0.001).

Among groups fed on the KD, the ANOVA showed an effect of the variable Week (F (5, 145) = 32.346; *p* < 0.001), and the interaction Week × Stress (F (10, 145) = 2.223; *p* < 0.019). Mice progressively reduced their levels of βHB from the 1st to the 4th week, tending to stabilize thereafter. The blood βHB levels during the 1st and 2nd week were higher in all groups compared to the subsequent weeks (for the 1st week: *p* < 0.05 with respect to the 2nd week and *p* < 0.001 with respect to the rest of the weeks; for the 2nd week *p* < 0.05 with respect to the 1st and 6th week and *p* < 0.001 with respect to the 3rd and 4th week, no differences with respect to the 5th week). Post hoc analyses revealed that the VSD-R group exhibited a more pronounced fluctuation in βHB levels, showing significantly lower βHB levels in the blood during the 3rd week compared to the 1st, 2nd, and 5th week (*p* < 0.01 with respect to the 1st and 5th week and *p* < 0.05 with respect to the 2nd week) (see Figure 3).

### 3.4. Corticosterone Levels

The ANOVA for the corticosterone levels during the first and fourth VSD sessions showed a significant effect of the variable Stress (F (2, 29) = 23.491; *p* < 0.001) and the interaction VSD session × Diet (F (1, 29) = 14.67; *p* < 0.001). The stressed groups, Resilient and Susceptible mice, showed higher corticosterone levels than the control non-stressed groups (*p* < 0.001 in both cases). Post hoc analyses revealed that the groups subjected to the KD exhibited elevated blood corticosterone levels following the first VSD compared to the groups maintained on a STD (*p* < 0.001). In addition, KD-fed groups exhibited higher blood corticosterone levels after the first VSD compared to the fourth VSD (*p* < 0.01). Conversely, STD-fed groups demonstrated increased blood corticosterone levels after the fourth VSD compared to the first VSD (*p* < 0.05) (see Figure 4).

### 3.5. Social Interaction Test (SIT)

The SIT ratio was determined by considering the duration that an experimental mouse spent in the interaction zone with a social target and dividing it by the duration it spent in the interaction zone without the target. The ANOVA for the SIT ratio showed a significant effect of the variable Stress (F (1, 57) = 13.128; *p* < 0.001). Post hoc analyses found that the VSD-Susceptible groups had a lower SIT ratio than the non-stressed and the VSD-Resilient groups (*p* < 0.001). No significant effects were found for the variable Diet or the interactions. However, the percentage of resilient or susceptible mice within the stressed groups was influenced by diet. A 62.5% of susceptible subjects and 37.5% of resilient subjects were identified in the KD-VSD group compared to 37.5% of susceptible subjects and 62.5% of resilient subjects in the STD-VSD group (see Figure 5).

### 3.6. Drinking in the Dark (DID)

The ANOVA for the g/kg of ethanol intake during the days of the DID test showed a significant effect of the variable Days (F (3, 159) = 9.384; *p* < 0.001), and the interaction Stress × Diet (F (2, 53) = 3.610; *p* < 0.034). There was a significant increase in ethanol intake on the 4th day in comparison with the 1st (*p* < 0.05), the 2nd (*p* < 0.001), and the 3rd (*p* < 0.01) day. Post hoc analyses also showed that susceptible female mice fed with the standard diet showed higher ethanol consumption than control and resilient females fed on the standard diet (*p* < 0.01 in all cases), and susceptible females fed on the KD (*p* < 0.05) (see Figure 6).

### 3.7. Oral Ethanol Self-Administration (SA)

Throughout the FR1, the ANOVA for the effective responses showed a significant effect of the variable Stress (F (2, 46) = 5.918; *p* < 0.005) and the interaction Stress × Diet (F (2, 46) = 3.478; *p* < 0.039). Post hoc analyses showed that the VSD-Susceptible females fed on STD performed a higher number of effective responses compared to the VSD-Resilient and non-stressed females fed with the same diet (*p* < 0.01) and to the VSD-Susceptible females fed on the KD (*p* < 0.05).

The ANOVA for the ethanol intake (g/kg) showed a significant effect of the variables Stress (F (2, 46) = 11.654; *p* < 0.001), Diet (F (1, 46) = 21.321; *p* < 0.001), and the interaction Stress × Diet (F (2, 46) = 6.380; *p* < 0.004). The STD-VSD groups (Resilient and Susceptible) exhibited significantly higher ethanol intake than the non-stressed females fed with the same diet (*p* < 0.01 and *p* < 0.001, respectively) and the resilient and susceptible females of the KD-VSD groups (*p* < 0.01 and *p* < 0.001, respectively).

During the PR, the ANOVA for the Breaking Point Value revealed a significant effect of the variables Stress (F (2, 46) = 4.252; *p* < 0.006), Diet (F (1, 46) = 11.309; *p* < 0.002), and the interaction Stress × Diet (F (2, 46) = 3.502; *p* < 0.013). The VSD-Susceptible females fed on STD had a more increased BPV than the STD-non-stressed females (*p* < 0.01) and VSD-Susceptible females fed on the KD (*p* < 0.001). No significant effects were found for the ethanol consumption or the number of effective responses in the PR (see Figure 7).

### 3.8. Gene Expression

#### 3.8.1. Striatum Samples

The ANOVA for DRD1 mRNA levels revealed a significant effect of the variable Stress (F (1, 28) = 4.821; *p* < 0.037) and the interaction Stress × Diet (F (1, 28) = 4.987; *p* < 0.034). Post hoc analyses showed that STD-VSD females had lower gene levels of DRD1 than STD-EXP females (*p* < 0.01) and KD-VSD (*p* < 0.01).

The ANOVA for the mRNA expression of the gene encoding the CB1 receptor (CNR1) found a significant effect of the variable Diet (F (1, 27) = 4.462; *p* < 0.044). The KD groups showed elevated expression levels of CRN1 compared to the STD groups (*p* < 0.05).

The ANOVA for the mRNA expression levels of the gene encoding the corticotropin-releasing factor receptor 1 (CRHR1) revealed a significant effect of the variable Diet (F (1, 28) = 9.656; *p* < 0.004). The KD groups showed lower expression levels of CRHR1 than the STD groups (*p* < 0.01).

The ANOVA for the mRNA expression levels of the gene encoding the mu-opioid receptor (OPRM) showed a significant effect of the variable Stress (F (1, 28) = 10.586; *p* < 0.003) and the interaction Stress × Diet (F (1, 28) = 4.473; *p* < 0.043). Post hoc analyses found that the STD-VSD group presented lower gene expression levels than the STD-EXP group (*p* < 0.001) and the KD-VSD group (*p* < 0.05).

Regarding neurotrophic factors, the ANOVA for the mRNA expression levels of the gene encoding brain-derived neurotrophic factor (BDNF) found a significant effect of the variable Stress (F (1, 27) = 13,278; *p* < 0.01), Diet (F (1, 27) = 6.344; *p* < 0.018), and the interaction Stress × Diet (F (1, 27) = 4.738; *p* < 0.038). Post hoc analyses showed that STD-VSD females had higher BDNF levels than STD-EXP females (*p* < 0.001) and KD-VSD (*p* < 0.01). On the other hand, the ANOVA for the mRNA expression levels of the gene encoding the tyrosine kinase receptor (TrkB), the target of BDNF, revealed a significant effect of the variable Diet (F (1, 27) = 5.002; *p* < 0.034). The KD-fed groups showed reduced levels of TrkB compared to STD-fed groups (*p* < 0.05).

Concerning interleukins, the ANOVA for the mRNA expression of the gene encoding IL-1β showed a significant effect of the interaction Stress × Diet (F (1, 27) = 6.619; *p* < 0.016). Post hoc analyses revealed that the KD-EXP group had higher IL-1β expression levels than the KD-VSD (*p* < 0.05) and STD-EXP (*p* < 0.05) groups.

The ANOVA for the mRNA expression levels of the gene encoding IL-10, an anti-inflammatory interleukin, showed a significant effect of the variable Stress (F (1, 28) = 4.575; *p* < 0.041), Diet (F (1, 28) = 11.656; *p* < 0.002), and the interaction Stress × Diet (F (1, 28) = 4.158; *p* < 0.05). Post hoc analyses found that the STD-VSD group had lower expression levels of IL-10 than the STD-EXP (*p* < 0.01) and the KD-VSD groups (*p* < 0.001).

The ANOVA for the gene encoding the dopamine 2 receptors (DRD2), ADORA1, ADORA2, IL-6, and the gene encoding the Toll-Like Receptor 4 (TLR4) expression levels did not show any significant effect on striatal tissue (see Figure 8).

#### 3.8.2. Hippocampus Samples

The ANOVA for DRD1 mRNA expression levels showed a significant effect of the interaction Stress × Diet (F (1, 28) = 7.392; *p* < 0.011). Post hoc analyses showed that the KD-VSD group increased their levels of DRD1 expression compared to the KD-EXP group (*p* < 0.05) and the STD-VSD group (*p* < 0.05).

The ANOVA for ADORA1 mRNA expression levels showed a significant effect of the variable Diet (F (1, 28) = 5.661; *p* < 0.024). The groups fed on the KD presented higher levels of ADORA1 expression compared to STD-fed groups (*p* < 0.05).

The ANOVA for CNR1 mRNA expression revealed a significant effect of the variable Diet (F (1, 28) = 21.939; *p* < 0.001). The KD-fed groups presented higher expression levels of CNR1 than STD-fed groups (*p* < 0.001).

The ANOVA for CRHR1 gene expression showed a significant effect of the variable Diet (F (1, 28) = 4.321; *p* < 0.047) and the interaction Stress × Diet (F (1, 28) = 4.240; *p* < 0.049). Post hoc analyses found that the STD-VSD group had decreased CRHR1 levels of expression compared to the STD-EXP group (*p* < 0.05). Post hoc analyses also showed that the KD-EXP group had lower levels of expression than the STD-EXP group (*p* < 0.01).

The ANOVA for OPRM mRNA levels found a significant effect of the variable Stress (F (1, 28) = 5.948; *p* > 0.021) and Diet (F (1, 28) = 29.942; *p* < 0.001). The VSD females presented increased levels of OPMR expression compared to non-stressed females (*p* < 0.05). In addition, the KD-fed groups showed higher levels of expression than STD-fed groups (*p* < 0.001). No effect of the interaction was found.

Concerning neurotrophic factors, the ANOVA for BDNF mRNA levels showed a significant effect of the variable Stress (F (1, 27) = 4.232; *p* < 0.049). The VSD groups decreased their expression of BDNF compared to the non-stressed groups (*p* < 0.05). No effects of the variable Diet or any interaction were found.

Regarding interleukins, the ANOVA for the IL-1β gene expression showed a significant effect of the variable Stress (F (1, 28) = 4.874; *p* < 0.036). The VSD females presented lower levels of IL-1β expression compared to the non-stressed females (*p* < 0.05). No effects of the variable Diet or any interaction were found.

For the gene expression levels of IL-6, the ANOVA found a significant effect of the variable Stress (F (1, 28) = 9.478; *p* < 0.005). The VSD females had decreased levels of IL-6 than non-stressed females (*p* < 0.01). No effects of the variable Diet or any interaction were found.

The ANOVA for IL-10 mRNA levels revealed a significant effect of the variable Stress (F (1, 27) = 4.524; *p* < 0.043), Diet (F (1, 27) = 6.575; *p* < 0.016), and the interaction Stress × Diet (F (1, 27) = 5.587; *p* < 0.026). Post hoc analyses found that the STD-VSD group had higher increased levels of expression compared to the STD-EXP (*p* < 0.01) and KD-VSD (*p* < 0.01) groups.

The ANOVA for TLR4 mRNA levels showed a significant effect of the variable Diet (F (1, 28) = 5.654; *p* < 0.024) and the interaction Stress × Diet (F (1, 28) = 6.192; *p* < 0.019). Post hoc analyses showed that the STD-VSD group had higher levels of TLR4 expression than the STD-EXP (*p* < 0.05) and KD-VSD (*p* < 0.01) groups (see Figure 9).

The ANOVA for DRD2, ADORA2, and TrkB expression levels did not show any significant effect on hippocampal tissue.

## 4. Discussion

The role of nutrition in substance-use disorders has been barely investigated, and to date, no study has assessed the influence of diet on the development of stress-induced drug intake. Previous reports have shown how the KD can restore microglial inflammatory activation and neuronal excitability in disorders like depression [46], and it has been associated with improvements in inflammatory and oxidative stress levels [47]. Previous studies had linked the KD with decreased ethanol consumption in male mice through changes in the expression of the ADORA 1 gene [35]; however, there are no studies relating the KD to increased ethanol consumption induced by social stress. Furthermore, in the field of substance abuse, most research has been conducted in male mice, with no existing research related to the influence of the KD on the development of addictive-like behaviors in female mice.

The findings of this study demonstrate that exposure to a KD during adolescence, despite enhancing categorization as a stress-susceptible phenotype in the SIT, effectively inhibits the long-term escalation of stress-induced ethanol consumption and the motivation to obtain it in the ethanol SA and DID procedures. The KD also blocked the stress-induced effect on the expression of the DRD1, OPRM, BDNF, and IL-10 genes in the striatum, and IL-10 and TLR4 in the hippocampus. Moreover, with a general effect, the KD decreased the expression of CRHR1 and TRKB genes in the striatum, increased the levels of ADORA1 and OPRM in the hippocampus, and increased CNR1 gene expression in both brain structures.

### 4.1. KD Increased βHB Levels, Body Weight, and Kcal Intake

As expected, exposure to a KD led to a notable increase in βHB levels, indicating the initiation of ketosis within the initial 24 h of KD exposure. The diminished availability of glucose as the primary energy source prompts the liver to generate ketone bodies, serving as the principal energy substrate. The levels of βHB have been established as a dependable biomarker for evaluating ketosis status in both mice and humans [48]. Our findings align with previous research showing heightened βHB levels following KD administration in mice [35,36,49] and rats [50]. However, while βHB levels in the KD-fed groups were significantly elevated compared to the STD-fed groups, these levels exhibited variability, experiencing a significant increase during the first week followed by a progressive reduction until stabilization at levels close to 1 mmol/L. These results mirror those reported by Gandía-Blanco et al. [35] in male mice, suggesting an adaptation of the organism to the state of ketosis through the regulation of βHB levels. Depending on the stress response phenotype (Resilient or Susceptible), we observed a significant increase in βHB levels in the resilient profile during the fifth week, coinciding with the initiation of the VSD procedure. The current body of evidence proposes that ketone bodies, particularly βHB, function as molecules in response to stress, coordinating an antioxidant defense program to uphold redox homeostasis when faced with environmental and metabolic challenges [51,52]. Therefore, the elevated levels of βHB in the resilient profile following exposure to a stressor may represent one of the mechanisms through which this profile responds to stress with reduced vulnerability.

Concerning the impact of the KD on body weight, previous studies reveal conflicting findings. In our study, a noteworthy increase in body weight was observed in the KD-fed groups compared to the STD controls, persisting for 2 weeks after the transition to the STD. These results align with prior investigations in male mice, where the KD induced a significant elevation in body weight compared to control groups [35,53]. However, other studies have reported no discernible impact of the KD on body weight, showing the KD-fed groups had comparable weights to control groups in both males [36,41,54] and females [55]. Moreover, some studies have indicated that the KD can result in a reduction in body weight compared to control groups in mice [40,56] and rats [49]. The varied outcomes observed may be attributed to differences in sex, strain, species, and the age at which KD administration started in the experiments. In most studies, KD administration began in adulthood, involving male C57BL/6J mice or rats, in contrast to our study where administration started during adolescence, in female OF1 mice. In the study by Ródenas-González et al. [41], using male mice of the same strain and beginning KD administration on the same PND as in our study, mice fed with the KD did not exhibit differences in body weight compared to those on a standard diet. This discrepant result may be attributed to sex-related differences, with adolescent female mice tending to accumulate more of the additional calories provided by the diet type compared to adolescent male mice. The discrepancy in results highlights the unknown mechanisms underlying the relationship between KD consumption and body weight, emphasizing the need for further investigation into the factors that may contribute to these divergent outcomes.

In terms of kcal intake, the groups fed the KD showed significantly higher kcal consumption compared to the groups fed the STD throughout the entire KD administration period. This difference can be attributed to the formulation of each diet. While the STD provides 2.9 kcal/g, the KD supplies 6.7 kcal/g. Consequently, even though both groups consumed the same quantity of food in grams, the KD-fed group obtained more than double the number of kcal per ration. These differences disappeared when all groups switched to the STD. Nevertheless, during the initial two weeks on the STD, the groups previously exposed to the KD exhibited a reduction in kcal consumed in comparison to the groups fed with the STD throughout the entire procedure, which could indicate the adaptation of the animals to their new diet. On the other hand, the stressed groups exhibited a lower kcal consumption during the weeks in which the VSD episodes took place, but they increased their consumption compared to the non-stressed groups two weeks after the conclusion of the defeat episodes. Animal studies have typically revealed heightened food consumption in the presence of diverse stressors. However, under high levels of stress, a more probable response appears to be reduced eating [57]. Moreover, stress-induced overconsumption of kilocalories has been primarily linked to females [57]. Stressed female mice in this study may have reduced their kcal consumption due to the direct experience of the stressful situation and regulated the long-term effects by increasing kcal intake. Finally, the overall decrease in kilocalorie consumption in the tenth week could be attributed to the implementation of the DID. During this period, mice spent between 2 and 4 h outside their cages with only access to 20% diluted ethanol. Ethanol (6.9 kcal/g) has a higher energy density than carbohydrates or proteins (4.1 kcal/g), being the second-highest energy-dense food after fat (9.1 kcal/g) [58,59]. Physical limitations to food access, coupled with satiety induced by ethanol intake, may have led to this overall decrease in food intake across all groups in the tenth week.

### 4.2. KD Increases the Percentage of Susceptible Female Mice According to the SIT Ratio

The SIT is widely employed to assess social avoidance behavior induced by exposure to social stress and to subsequently categorize animals into resilient or susceptible stress profiles. Previous findings have demonstrated that repeated exposure to social defeat stress in rodents leads, in a percentage of animals, to the development of a susceptible profile characterized by pronounced social avoidance associated with a constellation of behavioral and physiological changes resembling depressive and anxiety-like symptoms. However, another subgroup of animals does not develop social avoidance, referred to as the resilient profile [60]. Our results demonstrate that exposure to VSD induces both of these phenotypes, since we found 62.5% of resilient animals compared to 37.5% of susceptible animals among females fed on the STD. This high percentage of resilient subjects can be attributed to the difficulty in developing social avoidance in females due to their inherently low predisposition to be attacked by a conspecific of any sex. This finding is consistent with previous results where females exposed to VSD did not exhibit social avoidance [18].

However, focusing on the different diets administered, the KD increased the percentage of susceptible profile females compared to the STD. While animals exposed to VSD and fed with the STD showed over 60% of resilient subjects, in the groups fed with the KD, this percentage dropped to 37.5%, with 62.5% being susceptible subjects. This indicates that the KD increases the predisposition to social avoidance in female mice exposed to stress. Previous studies have found contrasting results, with the ketogenic diet-fed group showing a higher percentage of resilient subjects compared to those fed the STD [61]. However, these results were observed in males who were adults when the administration of the KD began. These results suggest sex differences, where the shift in energy source caused by the KD would result in vulnerability to stress in females, as opposed to males. On the other hand, it could indicate that the shift in energy source caused by the KD is protective in adult organisms when facing stress exposure. However, it might be detrimental in a still-developing organism such as an adolescent, enhancing vulnerability to the development of social avoidance. Further research is needed regarding the categorization into susceptible or resilient profiles induced by the KD, especially in females and at different developmental stages.

### 4.3. VSD Increased Ethanol Consumption and KD Blocked This Effect

Administration of the KD during adolescence until the last VSD encounter decreased long-lasting ethanol consumption measured by the DID test and the SA procedure. Our experimental design aimed to assess the long-term effects of exposure to the KD during adolescence on the increase in ethanol consumption induced by VSD.

According to previous studies, female mice fed on the STD and exposed to VSD exhibited higher ethanol consumption than non-stressed mice [2]. In the DID test, this increase in consumption was observed only in the susceptible group, although in the ethanol SA procedure, both resilient and susceptible female mice showed higher consumption than non-stressed females. However, only the susceptible group exhibited a significantly higher number of correct responses and break-point values, demonstrating increased motivation for ethanol acquisition. These results partially resemble those found by Reguilón et al. [62] in males, where susceptible mice exhibited the highest ethanol consumption and demonstrated greater motivation to obtain it compared to resilient and non-stressed groups. Therefore, in male mice, the resilient profile for depressive-like behaviors also protects against increased ethanol consumption. Our results in females were not so clear. Although both resilient and susceptible stressed females increased ethanol consumption, higher motivation was only observed in susceptible females, which increased the number of effective responses to obtain the drug. The divergent results between males and females may indicate a sex difference in the level of protection exerted by the resilient profile on stress-induced alcohol consumption. The characterization of females as resilient or susceptible based on their social interaction may be less useful than in the case of males. Females in a stressful situation exhibit more social-approach behaviors than males, and even sexual motivation may be a confounding variable [63,64]. Additionally, the different nature of the stress experience, involving physical contact in the case of males and only vicarious experience in females, could participate in this partial resilient phenotype.

When evaluating the groups based on the administered diet, we found that exposure to the KD completely blocked the effect of VSD on ethanol consumption in both the resilient and susceptible females. In the DID test and ethanol SA, stressed females fed on the KD maintained consumption levels like non-stressed mice. The same occurred regarding their motivation to obtain ethanol. This indicates that a ketogenic state during adolescence and exposure to a repeated vicarious social stressor successfully blocks the long-term effect on ethanol consumption and motivation for the drug. Previous studies have demonstrated how the ketogenic diet reduces ethanol consumption and associated withdrawal symptoms in mice [35,37] and rats [39,40]. However, in all these studies, ethanol consumption occurred during the administration of the diet. Our results make a valuable contribution by showing that the relationship between the KD and ethanol consumption is more robust than previously thought, exerting its effect even on the long-term increase in consumption induced by stress long after the diet administration has concluded. Furthermore, we present the first results linking the ketogenic diet to ethanol consumption in females, which appear to align with the findings observed in males. All these results corroborate the therapeutic potential of the KD in terms of preventing the development of an ethanol-use disorder by reducing the potential of social stress.

### 4.4. KD Induces Long-Lasting Changes in Gene Expression

Although the brain tissue samples were obtained ten weeks after the end of stress and diet exposure, we observed long-lasting gene expression changes in response to VSD and KD exposure during adolescence. To better understand these changes, the effects observed on gene expression were divided into two main categories: alterations induced by the KD and alterations induced by VSD that the KD successfully blocked. In this study, we analyzed samples of striatum and hippocampus tissue, structures closely associated with the addictive process and stress response [65,66,67,68,69].

As we have previously reported for males, no significant differences were found in the striatum for the ADORA1 gene expression in female mice fed with a KD and exposed to ethanol [35]. However, we observed an increased expression of ADORA1 in the hippocampus in all the groups fed with the KD. This increase could be explained by the long-term effect of the exposure of these groups to a ketogenic metabolism and its relationship with adenosine. Adenosine increases during ketogenic metabolism and induces a tonic inhibition of neuronal excitability through A1 receptors in various brain regions, including the hippocampus [70]. This inhibition significantly impacts both basal synaptic activity and neuronal plasticity [70], potentially explaining our results. Neuronal inhibition induced by adenosine could diminish circuit overexcitation in response to stressors and the subsequent reinforcing effect of ethanol, thereby preventing the development of consumptive behavior. However, we did not observe an increase in the expression of the ADORA 2 gene as found by Blanco-Gandia et al. [35]. These differences could be due to sex, where male mice may exhibit greater vulnerability to changes in the expression of this gene induced by ethanol consumption compared to females. On the other hand, they could also be due to differences in the ketotic state during ethanol consumption presented in both studies. In our study, ethanol consumption assessment was conducted three weeks after female mice ceased KD consumption, whereas in Blanco-Gandia et al. [35], male mice consumed the KD during ethanol exposure. This suggests that the changes observed in Blanco-Gandia et al. [35] could be due to an interaction between diet administration and ethanol consumption, which disappears upon eliminating the ketosis state.

The KD increased gene expression of CNR1 in the striatum and hippocampus. Previous studies have not found differences in CNR1 gene expression between animals fed a KD and a standard diet following ethanol exposure in the striatum [35]. However, our results may be due to differences in sex, age, exposure to ethanol, and diet administration protocol. On the other hand, Gamelin et al. [71] observed that high-fat diets increased CNR1 expression in the hippocampus, which supports our findings). These results, along with those found for ADORA1 and DRD1 gene expression, suggest that inducing a ketosis state during adolescence leads to long-term changes in systems that modulate the responsiveness of neural circuits to stressors and the consequent reinforcing effect of ethanol.

In general, the KD reduced levels of CRHR1 gene expression in the striatum and hippocampus. The CRHR1 receptor plays a role in the activation of the HPA/HPI axis and anxiety-related behaviors [72]. Neuronal CRH activity is modulated by GABAergic inhibition, with GABA playing a key role in the regulation of the HPA/HPI axis [73]. Previous studies have demonstrated how β-hydroxybutyric acid produced by the ketogenic diet accumulates cerebral GABA and increases the GABA/glutamate ratio [74] a phenomenon that could explain the reduced levels of CRHR1 gene expression in the KD-fed groups in our study.

Regarding the specific effects induced by VSD, which were effectively counteracted by the KD, vicarious stress reduces the expression levels of DRD1 in the striatum in the STD-VSD group as previously described for SD in males [75]. However, this effect is not observed in the KD-VSD group. In the hippocampus, although VSD did not change the expression of DRD1, the KD increased the expression of the DRD1 gene in this region in stressed females. Montagud-Romero et al. [15] also observed a decrease in DRD1 levels in the hippocampus after the fourth SD and 3 weeks later in male mice. Despite methodological differences, these results suggest that VSD downregulates DRD1, and our present results confirm that the KD can counteract this effect.

Although previous studies associate ethanol exposure with an increase in the expression of DRD1 [76], we found that the group with the highest ethanol consumption (VSD-STD) showed the lowest expression levels of the DRD1 gene, an effect blocked by KD exposure. Therefore, the KD restores or even increases the gene expression of DRD1 after ethanol consumption, as we have previously reported for male mice [35]. The long-term blockade of the decrease in DRD1 expression induced by VSD in ketogenic groups could be explained by the effect of the KD on adenosine receptors and their interaction with dopaminergic receptors. Adenosine and dopamine receptors form heterodimers (A1-D1 and A2-D2) that interact antagonistically [77], with the A1 receptor exhibiting an inhibitory function on dopamine signaling via the D1 receptor [78].

VSD reduced OPRM gene expression levels in the striatum, an effect also blocked by the KD. The OPRM gene is associated with both ethanol and stress responses, showing downregulation after ethanol intake [79] but contrasting results in response to stress [80]. Although some studies reported an increase in OPRM expression after stress, our results are consistent with findings from Rodriguez-Arias et al. [81], which showed that SD in adolescent male mice decreased OPRM gene expression in the nucleus accumbens. Therefore, we observed a downregulation of OPRM gene expression in the VSD-STD group, which also had the highest ethanol consumption. For the ketogenic diet-fed groups, the upregulation of the OPRM gene in the hippocampus could be associated with the reduced tolerance to opioid agonists reported by Beltran et al. [82] in mice fed with a KD compared to those fed with a high-fat diet and a standard diet.

VSD increased the expression of the BDNF gene, an effect blocked by KD exposure in the striatum. Conversely, VSD reduced BDNF expression levels in the hippocampus, with no effects observed from the KD. Regarding TrkB gene expression in the striatum, the KD decreased its expression, with no differences observed in the hippocampus. BDNF levels in response to stress induce different alterations depending on the brain region they act upon. Thus, in the hippocampus, BDNF exhibits a resilience effect to stress; however, it presents the opposite effect in the nucleus accumbens (NAc), where it acts as a susceptibility factor to stress [83,84]. Miyanishi et al. [84] determined that BDNF mediates sensitivity to social defeat, finding that its expression levels in the striatum negatively correlated with the time spent in the interaction zone in mice exposed to social defeat. These data align with our results, wherein the group exposed to VSD and fed a standard diet exhibited significantly higher levels of BDNF expression in the striatum compared to the other groups. In the hippocampus, previous studies have found that stress increases BDNF immediately after exposure but reduces it in the long term [85,86], corroborating our results where only the stressed groups exhibited lower levels of BDNF in the hippocampus. On the other hand, evidence points to the glycolysis inhibitor 2-DG mimicking the effects of the KD, leading to a reported decrease in brain-derived BDNF expression and its primary receptor, TrkB [87,88,89]. This finding corroborates our results, wherein only the groups previously fed with the KD exhibited a decrease in TrkB gene expression in the striatum. Additionally, this reduction in TrkB expression could explain the protective effect of the KD on the increase in VSD-induced BDNF levels in the striatum, diminishing the response to BDNF.

Regarding the neuroinflammatory response, VSD altered related markers, and the KD managed to block several of these alterations. VSD reduces the gene expression of the proinflammatory cytokines IL-1β and IL-6 in the hippocampus. Exposure to SD has been shown to increase the expression of IL-1β and IL-6 in this structure [90,91]. The reduction in IL-1β and IL-6 levels observed in the stressed groups in our study may represent a long-term effect of VSD exposure. Initially, this exposure might have heightened proinflammatory cytokine levels, followed by the subsequent downregulation of their expression. For the anti-inflammatory cytokine IL-10, VSD reduced its gene expression levels in the striatum and increased them in the hippocampus, with the KD blocking both effects. Previous studies have demonstrated that stress exposure reduces IL-10 expression in the striatum and hippocampus [92]; hence, our hippocampal results could be explained by the increased ethanol consumption [93,94] by the STD-VSD group. On the other hand, the blockade of VSD-induced effects on IL-10 by the KD could be related to the increase in brain levels of IL-10 produced by this diet [95,96]. Lastly, VSD increased TLR4 gene expression levels in the hippocampus, with the KD blocking this effect. Previous studies have shown higher levels of TLR4 gene expression in the hippocampus in rats previously exposed to stress [97], thus corroborating our findings. The blockade of this effect mediated by the KD could be due to the inhibition of the TLR4/MyD88/NFκB/NLRP3 pathway. Our results confirmed that VSD increased TLR4 gene expression levels in the hippocampus, an effect that was blocked by the KD. Previous studies have demonstrated elevated levels of TLR4 gene expression in the hippocampus of rats previously exposed to stress [97], which aligns with our findings. The inhibition of this effect by the KD might be attributed to the suppression of the TLR4/MyD88/NFκB/NLRP3 pathway. βHB has been linked to a reduction in the pro-inflammatory interleukin IL-1β via inhibition of the NLRP3 inflammasome, which regulates the release of pro-inflammatory cytokines. It has also been shown to inhibit the processing of this cytokine in response to the TLR4 pathogen-associated molecular pattern. The modulation of this pathway by the KD could underlie its beneficial effects on neuroinflammation [98,99,100]. Table 3 summaries the results of the gene expression in our study.

## 5. Conclusions

Exposure to a KD during adolescence and VSD experience effectively blocked the increase in consumption and motivation for ethanol in adulthood in female mice (see Figure 10). During adolescence, the brain undergoes development and is particularly vulnerable to the influence of external factors. Based on the results obtained in the gene expression study, the KD seems to, on one hand, increase the activation of inhibitory neurotransmission systems, such as the ADORA1 and CNR1 receptors. This could result in a reduction in neuronal overexcitation, which may induce lower organism responsiveness to stressors and the consequent alterations they produce, playing a protective role against the vulnerability to increased ethanol consumption in later stages. On the other hand, its antioxidant and anti-inflammatory effects could directly influence the severity of alterations induced by stress exposure, reducing their negative consequences. Additionally, these dietary-induced changes continue to manifest long-term.

Our study suggests a therapeutic potential for the KD or exposure to a ketogenic state in the development of stress-induced ethanol consumption disorder in females. However, the results linking this dietary intervention to ethanol consumption are very limited, and further research is needed to determine its validity as a therapeutic tool.

## Figures and Tables

**Figure 1 nutrients-16-02814-f001:**
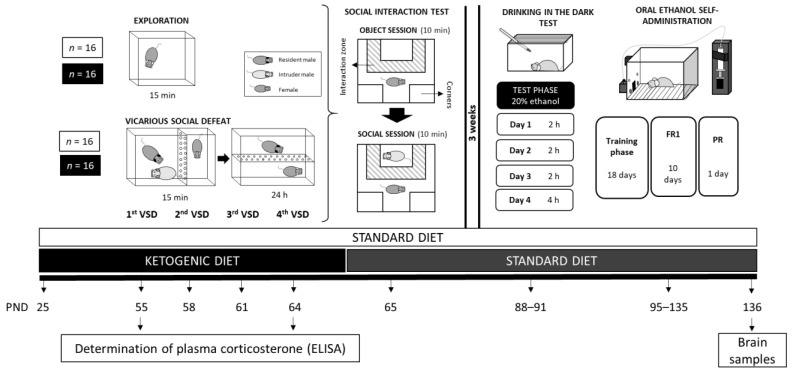
Experimental design.

**Figure 2 nutrients-16-02814-f002:**
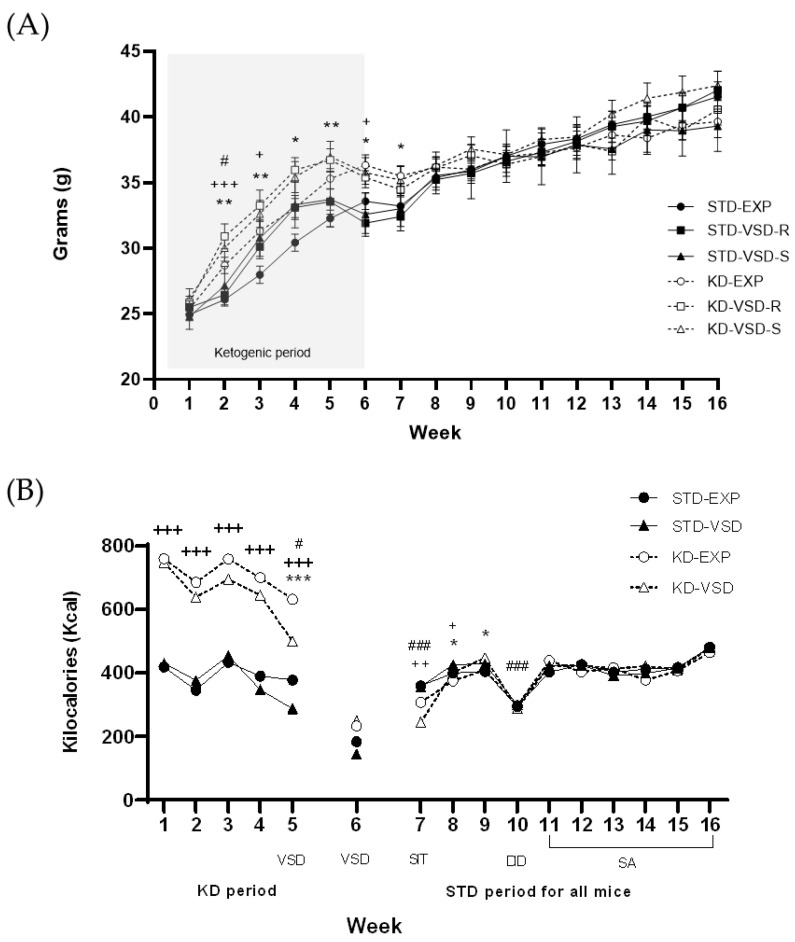
Weekly body weight and kilocalories (kcal) intake: (**A**) Weekly body weight. Data represented the body weight average per group measured weekly. The shaded area refers to the weeks in which the KD was maintained. Vertical lines show ± SEM. ** *p* < 0.01 and * *p* < 0.05 significant difference in KD-EXP groups with respect to the STD-EXP group. +++ *p* < 0.001 and + *p* < 0.05 significant difference in KD-VSD-R group with respect to the STD-VSD-R group. # *p* < 0.05 significant differences in KD-VSD-S group with respect to the STD-VSD-S group. (**B**) Weekly kcal intake. Data are represented as the average kcal intake per group measured weekly. Vertical lines show ±SEM. # *p* < 0.05 and ### *p* < 0.001 significant difference with respect to the rest of the weeks in each period. *** *p* < 0.001 and * *p* < 0.05 significant difference with respect to non-stressed females. +++ *p* < 0.001, ++ *p* < 0.01 and + *p* < 0.05 significant difference with respect to STD females.

**Figure 3 nutrients-16-02814-f003:**
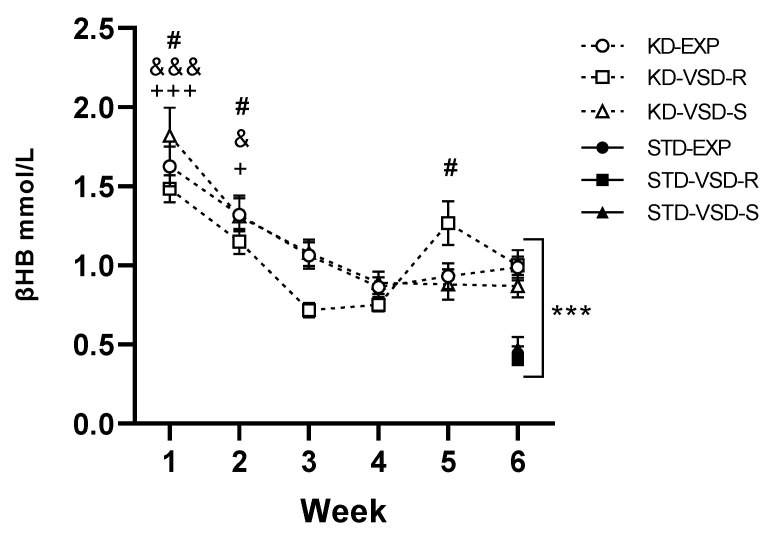
Weekly and β-hydroxybutyrate blood levels. Data are represented as the average of βHB plasma levels of KD groups measured weekly (black squares). Vertical lines show ±SEM. *** *p* < 0.001 significant differences with respect to the STD groups. +++ *p* < 0.001 and + *p* < 0.05 significant differences in the KD-EXP group with respect to the other weeks. # *p* < 0.05 significant differences in the KD-VSD-R groups with respect to the 3rd and 4th weeks. &&& *p* < 0.001 and & *p* < 0.05 significant differences in the KD-VSD-S group with respect to the other weeks. The black figures show the βHB plasma levels average of STD groups measured in the 6th week.

**Figure 4 nutrients-16-02814-f004:**
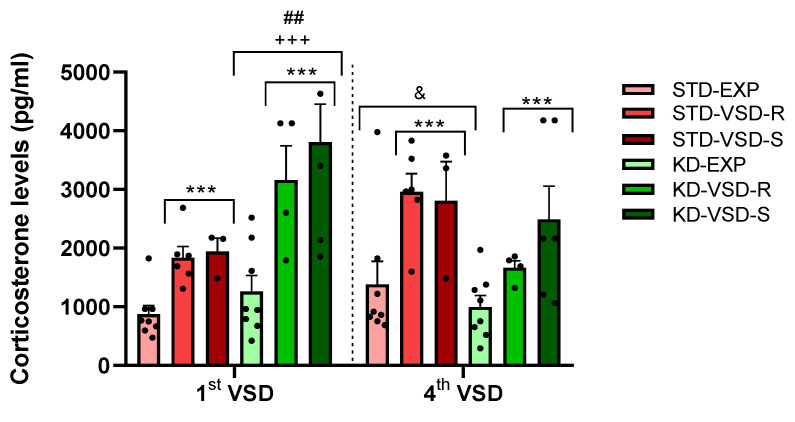
Corticosterone plasma levels after the 1st and 4th VSD sessions. The bars represent the corticosterone levels and the vertical lines show ±SEM. *** *p* < 0.001 significant differences with respect to the non-stressed groups. +++ *p* < 0.001 significant differences with respect to the STD groups. ## *p* < 0.05 significant differences with respect to the 4th VSD. & *p* < 0.05 significant differences with respect to the 1st VSD.

**Figure 5 nutrients-16-02814-f005:**
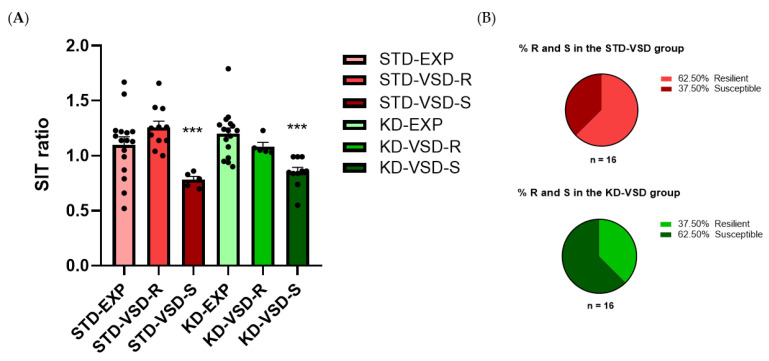
Social Interaction Test (SIT): (**A**) VSD and KD effect on the SIT ratio in female mice. The bars represent the ratio of the SIT and the vertical lines show ±SEM. Values >1 indicate preference for social interaction, and <1 indicates social avoidance. *** *p* < 0.001 significant difference with respect to the non-stressed and VSD-R groups. (**B**) KD increased the number of susceptible profiles in female mice. Data are represented as the percentage of resilient and susceptible subjects within the KD and STD-fed groups. The light gray portion represents the percentage of Resilient subjects (score > 1 on the SIT ratio), while the dark gray portion represents the percentage of susceptible subjects (score < 1 on the SIT ratio).

**Figure 6 nutrients-16-02814-f006:**
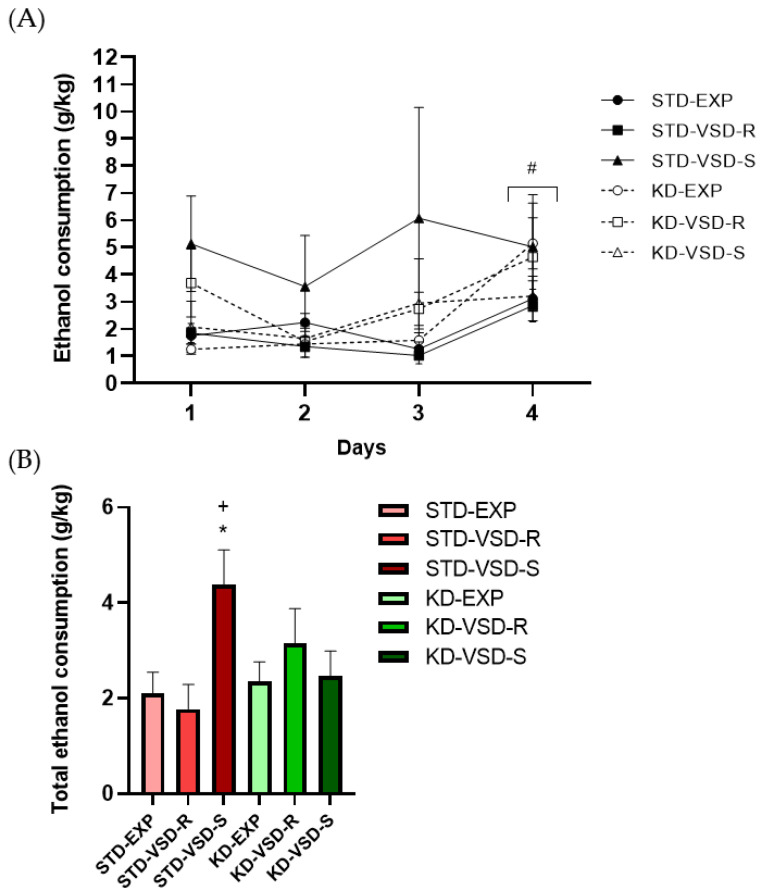
Drinking in the dark (DID) Test: (**A**) Female mice increase ethanol consumption with increased exposure time. Data are represented as the average of ethanol consumption per group during each test day. Vertical lines show ±SEM. # *p* < 0.05 significant differences in ethanol consumption on the fourth day with respect to the other three days. (**B**) Susceptible profile altered the ethanol consumption during the DID test and KD blocks this effect. The bars show the average total ethanol consumption per group. Vertical lines show ±SEM. * *p* < 0.05 significant differences with respect to non-stressed and VSD-Resilient females fed on STD. + *p* < 0.05 significant differences with respect to VSD-Susceptible females fed on KD.

**Figure 7 nutrients-16-02814-f007:**
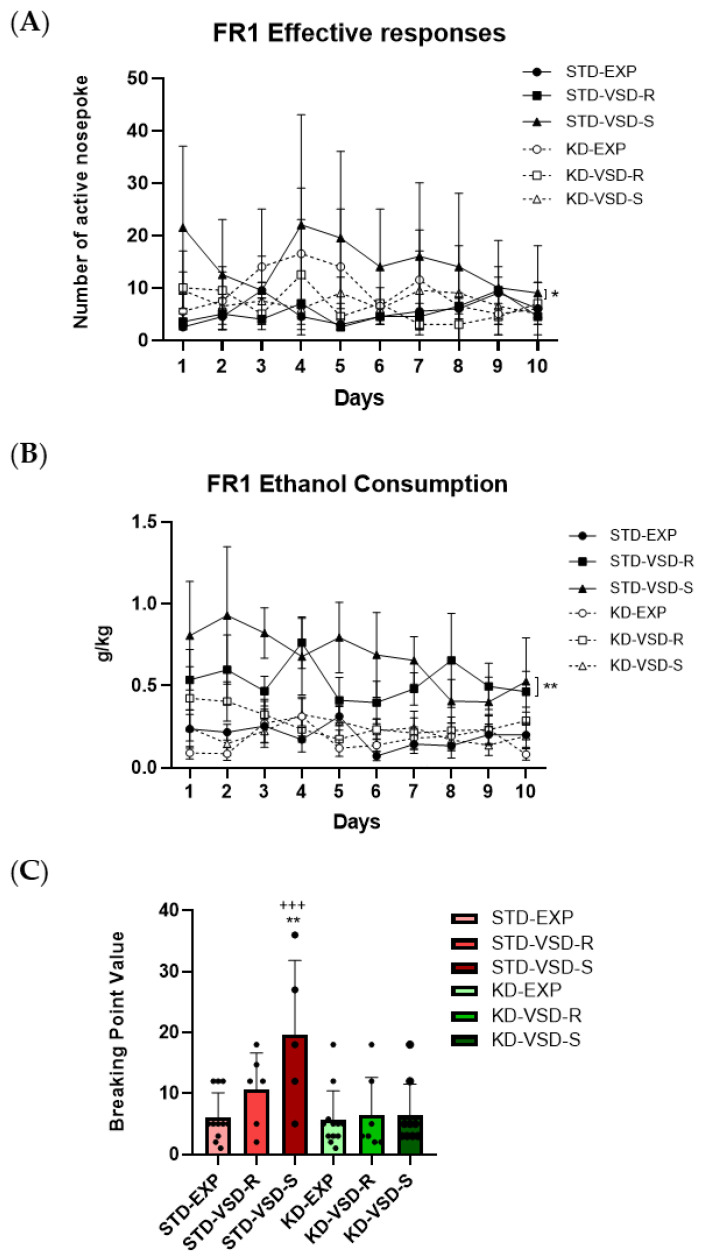
Oral ethanol self-administration (SA) procedure: (**A**) Susceptible profile and KD affected the number of effective responses during the FR1. Data are represented as the number of effective responses per group and vertical lines show ±SEM. * *p* < 0.05 significant differences in the STD-VSD-S group with respect to the rest of the groups. (**B**) VSD and KD affected ethanol consumption during FR1. Data are represented as the average of ethanol consumption per group and vertical lines show ±SEM. ** *p* < 0.01 significant differences in STD-VSD (Resilient and Susceptible) group with respect to the rest of the groups. (**C**) VSD and KD altered the Breaking Point Value. Data are represented as the average of BP value per group and vertical lines show ±SEM. ** *p* < 0.01 significant differences with respect to non-stressed females fed on STD. +++ *p* < 0.001 significant differences with respect to the KD-VSD-S group.

**Figure 8 nutrients-16-02814-f008:**
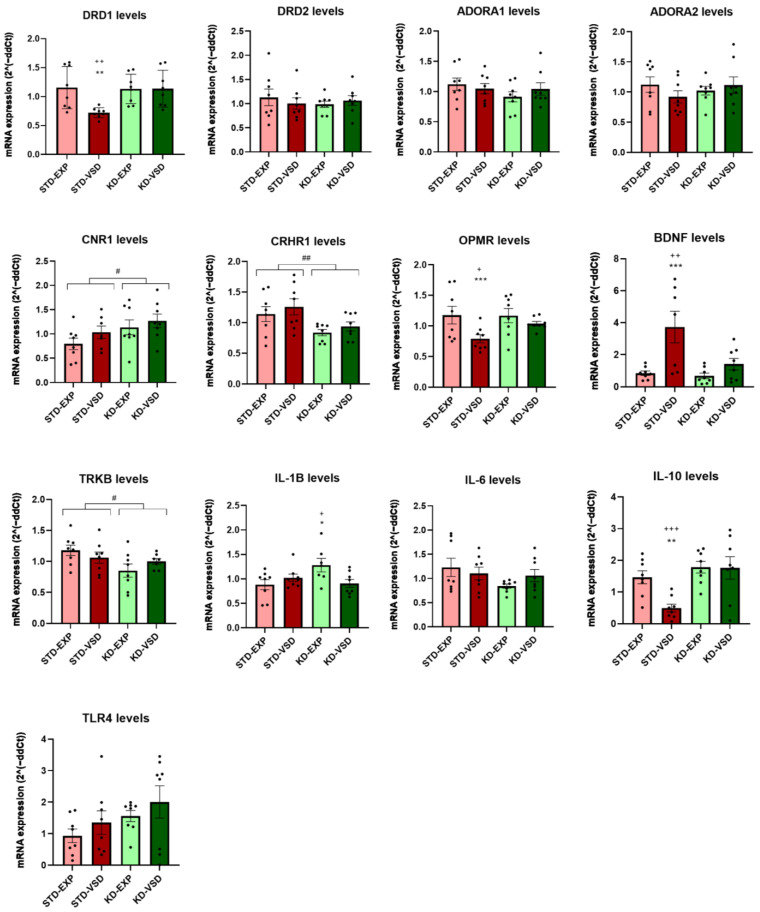
Gene expression in the striatum. The bars represent the mRNA expression levels and the vertical lines show ±SEM. *** *p* < 0.001; ** *p* < 0.01; * *p* < 0.05 significant difference with respect to the STD-EXP group. +++ *p* < 0.001; ++ *p* < 0.01; + *p* < 0.05 significant difference with respect to the KD-VSD group. ## *p* < 0.01; # *p* < 0.05 significant difference with respect to STD groups.

**Figure 9 nutrients-16-02814-f009:**
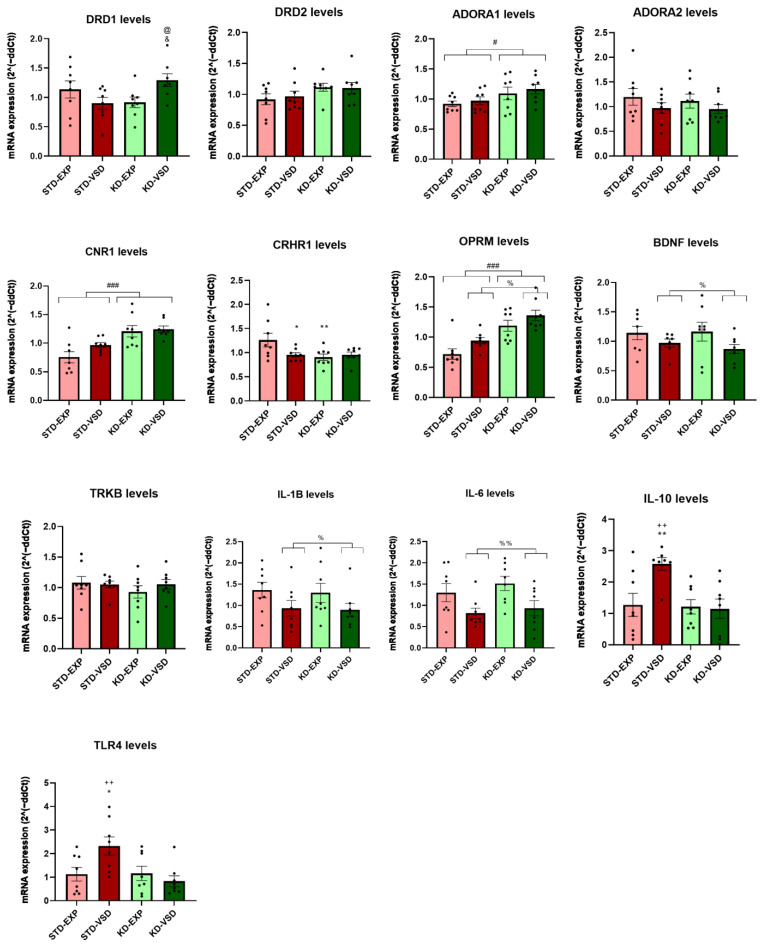
Gene expression in the hippocampus. The bars represent the mRNA expression levels and the vertical lines show ±SEM. * *p* < 0.05; ** *p* < 0.01 significant difference with respect to the STD-EXP group. ++ *p* < 0.01 significant differences with respect to the KD-VSD group. @ *p* < 0.05 significant difference with respect to the STD-VSD group. & *p* < 0.05 significant difference with respect to KD-EXP group. # *p* < 0.05; ### *p* < 0.001 significant differences with respect to STD groups. % *p* < 0.05; %% *p* < 0.01 significant differences with respect to non-stressed groups.

**Figure 10 nutrients-16-02814-f010:**
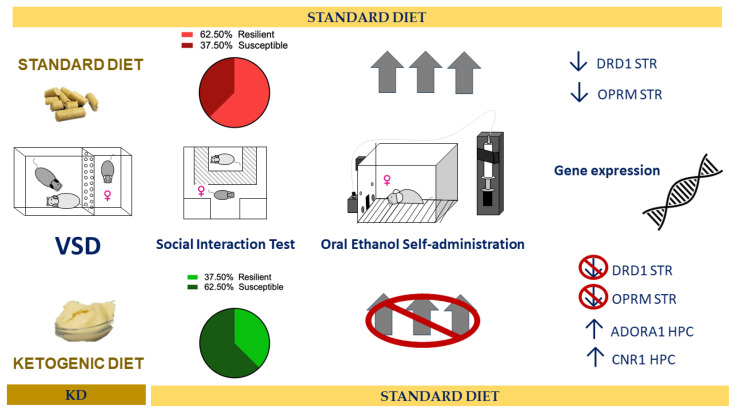
Summary of the main results.

**Table 1 nutrients-16-02814-t001:** Assay codes of the primers used for the RT-qPCR of genes.

Gene	Assay Codes
DrD1	Mm02620146
DrD2	Mm00438545
ADORA1	Mm01308023
ADORA2	Mm00802075
Cnr1	Mm01212171
Crhr1	Mm00432670
Oprm1	Mm01188089
β-Glucuronidase	Mm00446953

**Table 2 nutrients-16-02814-t002:** Nucleotide sequences of the primers used for the RT-qPCR of genes.

Gene	Primer Sequences (5′ to 3′)
IL-1β	F: GACCCCAAAAGATGAAGGGCT R: TGTGCTGCTGCGAGATTTGA
IL-6	F: AAGCCAGAGTCCTTCAGAGAGA R: TCTTGGTCCTTAGCCACTCCT
IL-10	F: TACCTGGTAGAAGTGATGCC R: CATCATGTATGCTTCTATGC
TLR4	F: TGCCTCTCTTGCATCTGGCTGG R: CTGTCAGTACCAAGGTTGAGAGCTGG
BDNF	F: CGCCAAGGTGGATGAGAGTT R: TTCGGCTTTGCTCAGTGGAT
TRKB	F: CCACGGATGTTGCTGACCAAAG R: GCCAAACTTGGAATGTCTCGCC
Cyclophilin A	F: GTCTCCTTCGAGCTGTTTGC R: GATGCCAGGACCTGTATGCT

**Table 3 nutrients-16-02814-t003:** Summary of results in the gene expression study.

Gene	Striatum	Hippocampus
ADORA 1	No differences	KD increased expression levels
DRD1	VSD reduced expression levels and KD blocks this effect	VSD and KD increased expression levels
CNR1	KD increased expression levels	KD increased expression levels
OPRM	VSD reduced expression levels and KD blocks this effect	VSD and KD increased expression levels
CRHR1	KD reduced expression levels	KD reduced expression levels
BDNF	VSD increased expression levels and KD blocks this effect	VSD reduced expression levels
TrkB	KD decreased expression levels	No differences
IL-1β	KD without VSD exposure increased expression levels	VSD reduced expression levels
IL-6	No differences	VSD reduced expression levels
IL-10	VSD reduced expression levels and KD blocks this effect	VSD increased expression levels and KD blocks this effect
TLR4	No differences	VSD increased expression levels and KD blocks this effect

## Data Availability

The raw data supporting the conclusions of this article will be made available by the authors on request.

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
