# Peer review of "Effects of Ketogenic Diet on Increased Ethanol Consumption Induced by Social Stress in Female Mice"

_nutrients, 2024, doi:10.3390/nu16172814_

Round 1
Reviewer 1 Report
Comments and Suggestions for Authors
The study conducted by Torres-Rubio et al. evaluated the effect of a ketogenic diet on increasing ethanol intake due to vicarious social defeat (VSD) in female 17 mice. Although the study is well conducted and original, there are some points that need to be improved.
To make it easier for the reader, it is important to describe the meaning of the abbreviations mentioned for the first time in the text, for example: CX3CL1, CXCL12, DRD1, among others;
It would be interesting for the authors to measure the cytokines using the ELISA method instead of RT-PCR, since they were able to obtain data on the protein formed;
The authors should describe in the statistical analysis which post hoc test was used;
Once activated, TLR4 can increase the levels of pro-inflammatory cytokines. It is interesting for the authors to discuss this mechanism;
It is important to include a figure that summarizes all the findings of the study.
Author Response
Reviewer 1
To make it easier for the reader, it is important to describe the meaning of the abbreviations mentioned for the first time in the text, for example: CX3CL1, CXCL12, DRD1, among others
These abbreviations have been introduced in the manuscript:
Dopamine receptor D1 and D2 (DrD1, DrD2), adenosine A1 and A2 receptor (ADORA1, ADORA2), Cannabinoid receptor 1 (CNR1), Corticotropin releasing hormone receptor 1 (CRHR1), Opioid receptor mu-1 (OPRM), Brain Derived Neurotrophic Factor (BDNF), Tropomyosin receptor kinase B (TrkB), Interleukin-6 (IL-6), Interleukin-1β (IL-1β), Interleukin-10 (IL-10), Toll-like receptor-4 (TLR4)
It would be interesting for the authors to measure the cytokines using the ELISA method instead of RT-PCR, since they were able to obtain data on the protein formed;
While it would have been ideal to conduct the studies suggested by the reviewer, we unfortunately do not have the necessary samples to do so. Conducting these studies would require redoing the entire experiment to obtain additional samples, which is not feasible. We will take this into consideration for future publications.
Additionally, it is important to emphasize that brain samples were collected at the conclusion of the entire procedure, not immediately after the last VSD. As a result, alcohol exposure across all groups may influence cytokine and chemokine levels. Gene expression provides us with dynamic information about the changes that occur at the end of the experiment.
Gene expression data are highly valuable and offer different insights compared to the ELISA-based determination of protein content. The significance and validity of gene expression studies are underscored by the numerous publications in which we have exclusively presented this type of data, particularly in relation to the consumption of various drugs.
González-Portilla M, Mellado S, Montagud-Romero S, Rodríguez de Fonseca F, Pascual M, Rodríguez-Arias M. Oleoylethanolamide attenuates cocaine-primed reinstatement and alters dopaminergic gene expression in the striatum. Behav Brain Funct. 2023 May 24;19(1):8. doi: 10.1186/s12993-023-00210-1. PMID: 37226219; PMCID: PMC10207629.
Blanco-Gandía MDC, Ródenas-González F, Pascual M, Reguilón MD, Guerri C, Miñarro J, Rodríguez-Arias M. Ketogenic Diet Decreases Alcohol Intake in Adult Male Mice. Nutrients. 2021 Jun 24;13(7):2167. doi: 10.3390/nu13072167. PMID: 34202492; PMCID: PMC8308435.
Ródenas-González F, Blanco-Gandía MDC, Pascual M, Molari I, Guerri C, López JM, Rodríguez-Arias M. A limited and intermittent access to a high-fat diet modulates the effects of cocaine-induced reinstatement in the conditioned place preference in male and female mice. Psychopharmacology (Berl). 2021 Aug;238(8):2091-2103. doi: 10.1007/s00213-021-05834-7. Epub 2021 Mar 31. PMID: 33786639.
Gasparyan A, Navarrete F, Rodríguez-Arias M, Miñarro J, Manzanares J. Cannabidiol Modulates Behavioural and Gene Expression Alterations Induced by Spontaneous Cocaine Withdrawal. Neurotherapeutics. 2021 Jan;18(1):615-623. doi: 10.1007/s13311-020-00976-6. Epub 2020 Nov 23. PMID: 33230690; PMCID: PMC8116402.
The authors should describe in the statistical analysis which post hoc test was used;
In 2.4. Statistical analysis section we specified that Bonferroni tests were used for post hoc comparisons.
Once activated, TLR4 can increase the levels of pro-inflammatory cytokines. It is interesting for the authors to discuss this mechanism;
Numerous evidence highlights the critical role of neuroimmune signaling in normal brain development, particularly in the regulation of synaptic pruning, neurite outgrowth, myelination, and neurotransmission. However, when neuroimmune responses are excessively activated, they can lead to neuronal death and myelin damage, which may result in significant cognitive and behavioral impairments (Guerri and Pascual, 2010).
Type I IL-1R (IL-1RI) is a receptor predominantly expressed on astrocytes, particularly following injury, indicating a specific association with inflammatory responses. TLR4, a member of the IL-1R/TLR superfamily, is also linked to inflammatory responses and is expressed in primary astrocytes. TLR4 is activated by proinflammatory cytokines and LPS, playing a crucial role in host responses to injury and infection. Upon ligand binding, TLRs and IL-1Rs dimerize and undergo the necessary conformational changes to recruit downstream signaling molecules, including the adaptor molecule MyD88, IL-1R-associated kinase (IRAK), and TNFR-associated factor 6. This recruitment initiates the activation of downstream kinases, such as MAPKs, and stress-activated protein kinases (SAPK/JNK), as well as the transcription factors NF-κB and AP-1. The activation of these transcription factors leads to the expression of genes encoding cytokines and inflammatory mediators, including inducible nitric oxide synthase (iNOS) and cyclooxygenase-2 (COX-2). Concomitantly, cell damage and apoptosis are triggered through the activation of signaling pathways and inflammatory mediators associated with TLR4/IL-1RI, resulting in a highly inflammatory response in the CNS under conditions of injury, disease, or stress (see review Blanco et al., 2005).
Numerous studies have shown that alcohol activates the neuroinflammatory response through TLR4 receptors (Pascual et al., 2018). Similarly, the increase in neuroinflammatory response induced by social defeat is mediated by the activation of these receptors (Montagud-Romero et al., 2018).
Our results confirmed that VSD increased TLR4 gene expression levels in​ the ​hippocampus, with KD blocking this effect. Previous studies have shown higher levels of TLR4 gene expression in the hippocampus in rats previously exposed to stress (Timberlake et al., 2018), thus corroborating our findings. The blockade of this effect mediated by KD could be due to the inhibition of the TLR4/MyD88/NFκB/NLRP3 pathway. βHB has been associated with the reduction of the pro-inflammatory interleukin IL-1β through the inhibition of the NLRP3 inflammasome, which controls the release of pro-inflammatory cytokines. It has also been shown to inhibit the processing of this cytokine in response to the TLR4 pathogen-associated molecular pattern. The modulation of this pathway by KD could be the cause of its beneficial effects on neuroinflammation (Dilimulati et al., 2022; Sun et al., 2023; Youm et al., 2015).
The following information has been added to the discussion section:
Our results confirmed that VSD increased TLR4 gene expression levels in the hippocampus, an effect that was blocked by KD. Previous studies have demonstrated elevated levels of TLR4 gene expression in the hippocampus of rats previously exposed to stress [97], which aligns with our findings. The inhibition of this effect by KD might be attributed to the suppression of the TLR4/MyD88/NFκB/NLRP3 pathway. βHB has been linked to a reduction in the pro-inflammatory interleukin IL-1β via inhibition of the NLRP3 inflammasome, which regulates the release of pro-inflammatory cytokines. It has also been shown to inhibit the processing of this cytokine in response to the TLR4 pathogen-associated molecular pattern. The modulation of this pathway by KD could underlie its beneficial effects on neuroinflammation [98, 99, 100].
Guerri C, Pascual M. Mechanisms involved in the neurotoxic, cognitive, and neurobehavioral effects of alcohol consumption during adolescence. Alcohol. 2010 Feb;44(1):15-26. doi: 10.1016/j.alcohol.2009.10.003. PMID: 20113871.
Blanco AM, Vallés SL, Pascual M, Guerri C. Involvement of TLR4/type I IL-1 receptor signaling in the induction of inflammatory mediators and cell death induced by ethanol in cultured astrocytes. J Immunol. 2005 Nov 15;175(10):6893-9. doi: 10.4049/jimmunol.175.10.6893. PMID: 16272348.
Pascual M, Montesinos J, Guerri C. Role of the innate immune system in the neuropathological consequences induced by adolescent binge drinking. J Neurosci Res. 2018 May;96(5):765-780. doi: 10.1002/jnr.24203. Epub 2017 Dec 7. PMID: 29214654.
Montagud-Romero S, Reguilón MD, Pascual M, Blanco-Gandía MC, Guerri C, Miñarro J, Rodríguez-Arias M. Critical role of TLR4 in uncovering the increased rewarding effects of cocaine and ethanol induced by social defeat in male mice. Neuropharmacology. 2021 Jan;182:108368. doi: 10.1016/j.neuropharm.2020.108368. Epub 2020 Oct 24. PMID: 33132187.
Timberlake, M.; Prall, K.; Roy, B.; Dwivedi, Y. Unfolded protein response and associated alterations in toll-like receptor expression and interaction in the hippocampus of restraint rats. Psychoneuroendocrinology 2018, 89, 185-193. [https://doi.org/gdf6f9]
Dilimulati, D.; Zhang, F.; Shao, S.; Lv, T.; Lu, Q.; Cao, M.; Jin, Y.; Jia, F.; Zhang, X. Ketogenic Diet Modulates Neuroinflammation via Metabolites from Lactobacillus reuteri After Repetitive Mild Traumatic Brain Injury in Adolescent Mice. Cell Mol Neurobiol 2022, 43, 907-923. [https://doi.org/m6dk]
Sun, W.; Wang, Q.; Zhang, R.; Zhang, N. Ketogenic diet attenuates neuroinflammation and induces conversion of M1 microglia to M2 in an EAE model of multiple sclerosis by regulating the NF-κB/NLRP3 pathway and inhibiting HDAC3 and P2X7R activation. Food Funct. 2023, 14, 7247-7269. [https://doi.org/m6dm]
Youm, Y. H., Nguyen, K. Y., Grant, R. W., Goldberg, E. L., Bodogai, M., Kim, D., ... & Dixit, V. D. The ketone metabolite β-hydroxybutyrate blocks NLRP3 inflammasome–mediated inflammatory disease. Nature medicine, 2015, 21, 263-269. [https://doi.org/gf5c73]
Final del formulario
It is important to include a figure that summarizes all the findings of the study.
A new Figure 10 has been added to the Conclusion section

Reviewer 2 Report
Comments and Suggestions for Authors
This manuscript reports on the effects of KD on vicarious social defeat stress induced increase in ethanol drinking in female mice. The study is well designed and carefully carried out. The data were adequately analysed by the appropriate statistical tools and reported in a critical and reflective manner. The finding that KD attenuates the VSD stress induced ethanol drinking is an important one. There results expand upon the growing body of literature showing benefits of KD in psychiatric conditions. The extension of previous knowledge obtained in males to females is an important development. The study also points to potential mechanisms underlying the anti-stress effects KD through key neurotransmitter receptors and cytokines and their receptors. This is a significant addition to the field. It will be important in the future to follow up those leads with more mechanistically orientated studies.
I would be interested in reading about what the authors think regarding the potential pathways that may mediate the effects of KD. One obvious one is BHB, through GABA as mentioned in the text. How about the gut microbiome, interaction with BHB with receptors, cytokines, etc? It has been demonstrated that BHB as an epigenetic modifier affects BDNF gene expression. Any similar mechanisms?
Overall, this is a well written paper with detailed and well argued discussion, which covers the findings thoroughly and critically. I would consider the questions above as relatively minor issues that can be easily addressed in the revised version.
However, I want the authors to spell out abbreviations for receptors when first mentioned. It should not be expected from all readers to immediately be familiar with abbreviations used in a particular narrow field.
"literature indicates..." is used repeatedly and should be avoided or synonymous expressions should be used.
Comments on the Quality of English LanguageThe quality of English language use and style are good overall. However, the "literature indicates..." is used repeatedly and should be avoided or synonymous expressions should be used.
Author Response
I would be interested in reading about what the authors think regarding the potential pathways that may mediate the effects of KD. One obvious one is BHB, through GABA as mentioned in the text. How about the gut microbiome, interaction with BHB with receptors, cytokines, etc? It has been demonstrated that BHB as an epigenetic modifier affects BDNF gene expression. Any similar mechanisms?
Broadly speaking, the ketogenic diet (KD) exerts its primary effects by reducing inflammatory responses, mitigating oxidative stress through enhanced mitochondrial function, and diminishing neuronal hyperexcitability mediated by GABA and adenosine. These mechanisms collectively provide a compensatory response to the alterations induced by stress exposure and subsequent ethanol intake.
Regarding the gut microbiome, evidence suggests that the influence of the KD on GABA levels may be more closely associated with its capacity to regulate the balance between beneficial and harmful bacteria (Tang et al., 2021), rather than being directly linked to the ketogenic state or BHB levels (Hartman et al., 2007). Moreover, ketone bodies, such as beta-hydroxybutyrate, inhibit the proliferation of bifidobacteria, resulting in a reduction of pro-inflammatory intestinal Th17 cells (Ang et al., 2020). These effects may serve as protective factors against stress- and ethanol-induced disruptions within this axis, such as increased intestinal barrier permeability, thereby preventing dysregulation (Dinan & Cryan, 2012; Meroni et al., 2019).
Regarding adenosine, one of the key therapeutic benefits of the KD in epilepsy treatment is the enhanced activation of A1 adenosine receptors, potentially attributable to the diet's ability to decrease adenosine kinase, the primary enzyme responsible for adenosine metabolism (Masino et al., 2011). A1 adenosine receptors are highly expressed in the striatum and nucleus accumbens, regions involved in the brain's reinforcement circuitry and the reinforcing effects of substances such as alcohol. In these regions, adenosine receptors form antagonistic heterodimers with dopamine receptors (A1-D1 and A2-D2) (Franco et al., 2000). Consequently, the KD-induced upregulation of A1 adenosine receptor activity could inhibit D1 dopaminergic receptor activity in these areas, potentially leading to a diminished reinforcing effect of ethanol.
Additionally, BHB is implicated in the downregulation of pro-inflammatory interleukin IL-1β through the inhibition of the NLRP3 inflammasome (Youm et al., 2015). TLR4 is linked to inflammatory responses and is expressed in primary astrocytes. TLR4 is activated by proinflammatory cytokines and LPS, playing a crucial role in host responses to injury and infection. Upon ligand binding, TLRs and IL-1Rs dimerize and undergo the necessary conformational changes to recruit downstream signaling molecules, including the adaptor molecule MyD88, IL-1R-associated kinase (IRAK), and TNFR-associated factor 6. This recruitment initiates the activation of downstream kinases, such as MAPKs, and stress-activated protein kinases (SAPK/JNK), as well as the transcription factors NF-κB and AP-1. The activation of these transcription factors leads to the expression of genes encoding cytokines and inflammatory mediators, including inducible nitric oxide synthase (iNOS) and cyclooxygenase-2 (COX-2). Concomitantly, cell damage and apoptosis are triggered through the activation of signaling pathways and inflammatory mediators associated with TLR4/IL-1RI, resulting in a highly inflammatory response in the CNS under conditions of injury, disease, or stress (see review Blanco et al., 2005).
Numerous studies have shown that alcohol activates the neuroinflammatory response through TLR4 receptors (Pascual et al., 2018). Similarly, the increase in neuroinflammatory response induced by social defeat is mediated by the activation of these receptors (Montagud-Romero et al., 2018).
Our results confirmed that VSD increased TLR4 gene expression levels in the hippocampus, with KD blocking this effect. Previous studies have shown higher levels of TLR4 gene expression in the hippocampus in rats previously exposed to stress (Timberlake et al., 2018), thus corroborating our findings. The blockade of this effect mediated by KD could be due to the inhibition of the TLR4/MyD88/NFκB/NLRP3 pathway. βHB has been associated with the reduction of the pro-inflammatory interleukin IL-1β through the inhibition of the NLRP3 inflammasome, which controls the release of pro-inflammatory cytokines. It has also been shown to inhibit the processing of this cytokine in response to the TLR4 pathogen-associated molecular pattern. The modulation of this pathway by KD could be the cause of its beneficial effects on neuroinflammation (Dilimulati et al., 2022; Sun et al., 2023; Youm et al., 2015).
The following information has been added to the discussion section:
Our results confirmed that VSD increased TLR4 gene expression levels in the hippocampus, an effect that was blocked by KD. Previous studies have demonstrated elevated levels of TLR4 gene expression in the hippocampus of rats previously exposed to stress [97], which aligns with our findings. The inhibition of this effect by KD might be attributed to the suppression of the TLR4/MyD88/NFκB/NLRP3 pathway. βHB has been linked to a reduction in the pro-inflammatory interleukin IL-1β via inhibition of the NLRP3 inflammasome, which regulates the release of pro-inflammatory cytokines. It has also been shown to inhibit the processing of this cytokine in response to the TLR4 pathogen-associated molecular pattern. The modulation of this pathway by KD could underlie its beneficial effects on neuroinflammation [98, 99, 100].
Tang, Y., Wang, Q., & Liu, J. (2021). Microbiota-gut-brain axis: a novel potential target of ketogenic diet for epilepsy. Current opinion in pharmacology, 61, 36-41.
Hartman, A. L., Gasior, M., Vining, E. P., & Rogawski, M. A. (2007). The neuropharmacology of the ketogenic diet. Pediatric neurology, 36(5), 281-292.
Ang, Q. Y., Alexander, M., Newman, J. C., Tian, Y., Cai, J., Upadhyay, V., ... & Turnbaugh, P. J. (2020). Ketogenic diets alter the gut microbiome resulting in decreased intestinal Th17 cells. Cell, 181(6), 1263-1275.
Dinan, T. G., & Cryan, J. F. (2012). Regulation of the stress response by the gut microbiota: implications for psychoneuroendocrinology. Psychoneuroendocrinology, 37(9), 1369-1378.
Meroni, M., Longo, M., & Dongiovanni, P. (2019). Alcohol or gut microbiota: who is the guilty?. International journal of molecular sciences, 20(18), 4568.
Youm, Y. H., Nguyen, K. Y., Grant, R. W., Goldberg, E. L., Bodogai, M., Kim, D., ... & Dixit, V. D. (2015). The ketone metabolite β-hydroxybutyrate blocks NLRP3 inflammasome–mediated inflammatory disease. Nature medicine, 21(3), 263-269.
Masino, S. A., Li, T., Theofilas, P., Sandau, U. S., Ruskin, D. N., Fredholm, B. B., ... & Boison, D. (2011). A ketogenic diet suppresses seizures in mice through adenosine A 1 receptors. The Journal of clinical investigation, 121(7), 2679-2683.
Franco, R., Ferré, S., Agnati, L., Torvinen, M., Ginés, S., Hillion, J., ... & Fuxe, K. (2000). Evidence for adenosine/dopamine receptor interactions: indications for heteromerization. Neuropsychopharmacology, 23(4), S50-S59.
Timberlake, M.; Prall, K.; Roy, B.; Dwivedi, Y. Unfolded protein response and associated alterations in toll-like receptor expression and interaction in the hippocampus of restraint rats. Psychoneuroendocrinology 2018, 89, 185-193. [https://doi.org/gdf6f9]
Dilimulati, D.; Zhang, F.; Shao, S.; Lv, T.; Lu, Q.; Cao, M.; Jin, Y.; Jia, F.; Zhang, X. Ketogenic Diet Modulates Neuroinflammation via Metabolites from Lactobacillus reuteri After Repetitive Mild Traumatic Brain Injury in Adolescent Mice. Cell Mol Neurobiol 2022, 43, 907-923. [https://doi.org/m6dk]
Sun, W.; Wang, Q.; Zhang, R.; Zhang, N. Ketogenic diet attenuates neuroinflammation and induces conversion of M1 microglia to M2 in an EAE model of multiple sclerosis by regulating the NF-κB/NLRP3 pathway and inhibiting HDAC3 and P2X7R activation. Food Funct. 2023, 14, 7247-7269. [https://doi.org/m6dm]
However, I want the authors to spell out abbreviations for receptors when first mentioned. It should not be expected from all readers to immediately be familiar with abbreviations used in a particular narrow field.
These abbreviations have been introduced in the manuscript:
Dopamine receptor D1 and D2 (DrD1, DrD2), adenosine A1 and A2 receptor (ADORA1, ADORA2), Cannabinoid receptor 1 (CNR1), Corticotropin releasing hormone receptor 1 (CRHR1), Opioid receptor mu-1 (OPRM), Brain Derived Neurotrophic Factor (BDNF), Tropomyosin receptor kinase B (TrkB), Interleukin-6 (IL-6), Interleukin-1β (IL-1β), Interleukin-10 (IL-10), Toll-like receptor-4 (TLR4)
"literature indicates..." is used repeatedly and should be avoided or synonymous expressions should be used.
Comments on the Quality of English Language
The quality of English language use and style are good overall. However, the "literature indicates..." is used repeatedly and should be avoided or synonymous expressions should be used.
This sentence has been modified